# Counterfactual LLM-based Framework for Measuring Rhetorical Style

**Jingyi Qiu**
School of Information
University of Michigan, Ann Arbor
`jaqiu@umich.edu`

**Hong Chen**
School of Information
University of Michigan, Ann Arbor
`hongcc@umich.edu`

**Zongyi Li**
CSAIL
MIT
`zongyili@mit.edu`

## Abstract

The rise of AI has fueled growing concerns about "hype" in machine learning papers, yet a reliable way to quantify rhetorical style independently of substantive content has remained elusive. Because bold language can stem from either strong empirical results or mere rhetorical style, it is often difficult to distinguish between the two. To disentangle rhetorical style from substantive content, we introduce a counterfactual, LLM-based framework: multiple LLM rhetorical personas generate counterfactual writings from the same substantive content, an LLM judge compares them through pairwise evaluations, and the outcomes are aggregated using a Bradley–Terry model. Applying this method to 8,485 ICLR submissions sampled from 2017 to 2025, we generate more than 250,000 counterfactual writings and provide a large-scale quantification of rhetorical style in ML papers. We find that visionary framing significantly predicts downstream attention, including citations and media attention, even after controlling for peer-review evaluations. We also observe a sharp rise in rhetorical strength after 2023, and provide empirical evidence showing that this increase is largely driven by the adoption of LLM-based writing assistance. The reliability of our framework is validated by its robustness to the choice of personas and the high correlation between LLM judgments and human annotations. Our work demonstrates that LLMs can serve as instruments to measure and improve scientific evaluation.

## 1 Introduction

Machine learning is a field of extraordinary excitement: every year brings breakthroughs in models, applications, and benchmarks. Meanwhile, the ML community has become intensely competitive, with leading conferences now receiving tens of thousands of submissions (e.g., ICLR 2025 received over 10,000 submissions, and submissions to ICLR 2026 nearly doubled, reaching close to 20,000). In this crowded field, visibility is scarce, and authors, when framing their work in research papers, face strong incentives to emphasize and sometimes overstate the significance of their contributions (Smaldino & McElreath, 2016; McGreivy & Hakim, 2024).

This pattern has fueled concerns about hype, a rhetorical style that exaggerates novelty or impact (Lazarus et al., 2015; Thais, 2024). Concerns about hype are not merely stylistic: rhetorical framing is often suspected of influencing how papers are perceived in peer review and in attracting downstream attention (Boutron et al., 2014). Despite these concerns, there is no systematic and widely adopted framework for measuring rhetorical style in scientific writing. Addressing this measurement gap is critical. Prior studies document rising use of positive framing and promotional language (Vinkers et al., 2015; Peng et al., 2024; Millar et al., 2024), declining expressions of uncertainty (Yao et al., 2023), and even widespread misrepresentation in published research (Boutron & Ravaud, 2018; McGreivy & Hakim, 2024; Chen et al., 2025).

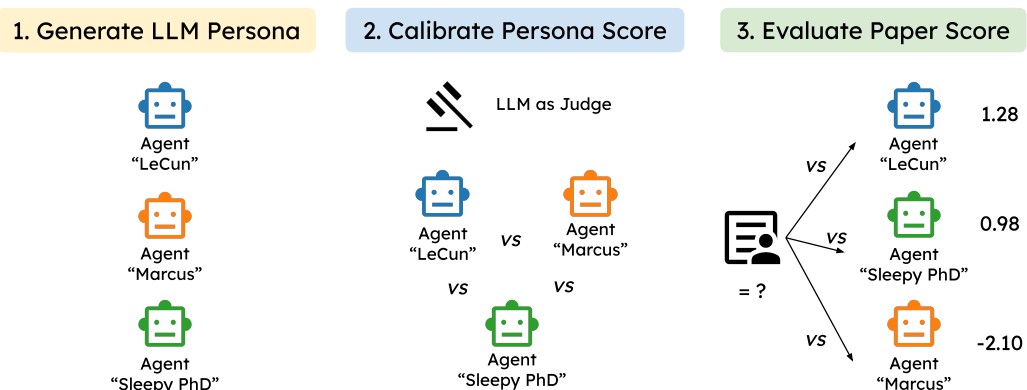

Figure 1: (1) LLM personas generate counterfactual writings in different rhetorical styles based on the same substantive content. (2) We calibrate the LLM personas' rhetorical scores via pairwise comparisons using an LLM judge. (3) We infer the rhetorical score of any query abstract by comparing it against the calibrated LLM persona panel.

Existing approaches primarily focus on subcomponents of rhetorical style inferred directly from the observed text alone: either by scoring promotional lexicons and indexes (Mishra et al., 2023; Peng et al., 2024; Gentzkow & Shapiro, 2010) or by training models on human-labeled constructs such as "sensationalism" or "uncertainty" (Prabhakaran et al., 2016; Pei & Jurgens, 2021; Wührl et al., 2024). These methods are effective at capturing surface-level tone, but because they rely on a single observed realization of the text, they conflate stylistic framing with the strength of the underlying evidence. Because bold language can stem from either strong empirical results or mere rhetorical style, it is often difficult to distinguish between the two. The central challenge, here, is to **distinguish a paper's presentation from its substantive contribution**.

We address this challenge by formalizing rhetorical style, conditioned on fixed substantive content, as a measurement problem. The core idea is simple: hold the substantive content $X$ (methods, experiments, results) constant, vary the descriptive language $Y$, and infer a latent variable $Z$ that captures rhetorical strength independently of substantive content. Just as benchmarks quantify empirical progress, this formulation provides a quantitative basis for analyzing rhetorical style and understanding how it shapes attention in ML research.

**Our Contributions.** We develop a counterfactual, LLM-based framework that measures rhetorical style independently of substantive content. As shown in Figure 1, we employ a diverse panel of LLM personas that generate counterfactual writings in different rhetorical styles from the same substantive content. These persona outputs are then compared in pairs by an LLM judge, and the pairwise comparison outcomes are aggregated with a Bradley–Terry model (Bradley & Terry, 1952) to assign each persona a calibrated rhetorical score. Finally, we infer the rhetorical score of any query abstract by positioning it relative to a calibrated panel of counterfactual persona abstracts derived from identical substantive content.

Conceptually, our design parallels a **counterfactual design** in causal inference, representing "what-if" alternative outcomes under different scenarios. We ask how the perception and evaluation of a paper would change if the rhetorical style were altered while the underlying substantive content remained the same. Each persona-generated abstract represents a counterfactual answer to a "what-if" question: "What would this abstract look like if the author had written it in a different style?" By coupling controlled generation with pairwise comparison under a classic statistical framework, we introduce a scalable and principled LLM-based instrument for measuring and improving scientific evaluation.

- **Methodology:** We introduce a counterfactual evaluation paradigm using a calibrated panel of LLM personas as controlled writers that generate counterfactual writings from identical substantive content, with pairwise judgments aggregated via a Bradley–Terry model.
- **Validation:** Unlike prior approaches that infer rhetorical strength directly from the observed text, our framework controls for substantive content, yields fine-grained scores, and is robust

to the choice of personas. We further validate the LLM-based judgments against human annotations.

- **Findings:** Applying our framework to 8,485 ICLR submissions from 2017 to 2025 and more than 250,000 generated persona writings and 250,000 pairwise comparisons, we provide large-scale evidence that visionary rhetorical style predicts citations and media attention, and that rhetorical strength has risen significantly since 2023. We further show that this increase is largely driven by the adoption of LLM-based writing assistance.

Our use of pairwise comparisons is closely related to preference-based learning methods such as Reinforcement Learning from Human Feedback (RLHF) and Direct Preference Optimization (DPO), which leverage pairwise preferences over multiple outputs generated from the same input to learn a reward model (RLHF) or directly optimize a policy (DPO) (Christiano et al., 2017; Ouyang et al., 2022; Rafailov et al., 2023). Related approaches in political science likewise use multiple texts describing the same event to infer latent attributes, such as political leaning (Carlson & Montgomery, 2017; Licht et al., 2025). While these methods typically use pairwise preferences for model training or policy optimization, our work adapts this paradigm for a distinct purpose: to create a calibrated measurement scale for a latent attribute (rhetorical style) by holding substantive content constant.

## 2 Problem Setting

### 2.1 Problem Formulation

We formalize rhetorical style as a latent variable in a generative model of research writing. Specifically, we denote the text of a paper as $Y$, which is influenced by its substantive content $X$ and a latent rhetorical style variable $Z$.

For simplicity, we focus on measuring rhetorical style in abstracts, since they are typically the first section readers encounter and contain the core articulation of a paper's contributions and significance. Accordingly, we define $Y$ as the abstract text, while $X$ refers to the substantive basis, such as preliminaries, methods, experiments, and results, which are comparatively objective and descriptive. The rhetorical style $Z$ is modeled as a latent continuous scalar capturing the strength of framing. Formally, $X \in \{1, \ldots, T\}^n$ and $Y \in \{1, \ldots, T\}^m$ are token sequences of finite length, and $Z \in \mathbb{R}$ is a one-dimensional hidden variable representing rhetorical strength.

We posit that the observed writing $Y$ is generated from a conditional distribution:

$$p(Y \mid X, Z) \tag{1}$$

where $X$ refers to the substantive content and $Z$ modulates the framing of the contribution. The objective of this work is to infer $Z$ independently of $X$, yielding a quantitative measure of rhetorical strength disentangled from substantive content.

Traditional approaches to measuring rhetorical style in scientific texts rely on direct linguistic features of $Y$, but such measures are confounded by $X$, since stronger results naturally justify stronger claims. Our approach instead targets $Z$, isolating rhetorical style conditional on fixed substantive content.

### 2.2 Rhetorical Strength

We assume that a total order exists in the space of latent rhetorical style $Z$, as $Z \in \mathbb{R}$. A higher value of $Z$ corresponds to a "stronger" rhetorical style (e.g., more visionary), while a lower value suggests a "weaker" rhetorical style (e.g., more conservative).

**Definition 2.1** (Stronger Rhetorical Style). *A **stronger rhetorical style** refers to a rhetorical style that broadly expands its implications. This involves generalizing its applicability, highlighting the significant challenges it addresses, and emphasizing its novelty and impact.*

**Definition 2.2** (Weaker Rhetorical Style). *A **weaker rhetorical style** refers to a rhetorical style that narrowly confines its implications, potentially by specifying its applicability and prerequisites, positioning it closely to previous work, and emphasizing its limitations and uncertainties.*

For any two rhetorical styles $z_1, z_2 \in Z$, if $z_1 > z_2$, then $z_1$ represents a stronger rhetorical style than $z_2$.

**Definition 2.3** (Order of Rhetorical Strength). *Given a fixed substantive basis $X = x$, for any two writings $y_1, y_2$ derived from $(x, z_1)$ and $(x, z_2)$, we say $y_1$ has a **stronger rhetorical style** than $y_2$ (denoted $y_1 \succ_x y_2$) if and only if $z_1 > z_2$. It implies $y_1$ presents the writing with greater assertiveness, broader scope, or higher proclaimed impact than $y_2$.*

In practice, judgments about whether $y_1$ is rhetorically stronger than $y_2$ can be noisy, affected by both uncertainty and observation error. We therefore model these comparisons using the Bradley–Terry model (Section 3.3).

While we acknowledge that rhetorical style is inherently multi-faceted, i.e., encompassing distinct dimensions such as claims of novelty, generalizability, and impact, we model it here as a single latent dimension of "strength." This simplification is a principled choice for this foundational work, as it allows us to first establish whether rhetorical framing can be cleanly disentangled from content. The framework itself is extensible, and future work could adapt the judging criteria to measure a multi-dimensional representation of rhetorical style.

## 3 COUNTERFACTUAL LLM-BASED FRAMEWORK

In this section, we introduce a counterfactual LLM-based framework for measuring latent rhetorical style. As shown in Figure 1, the method proceeds in two stages. First, we construct a panel of LLM persona writers with specified rhetorical tendencies and calibrate their rhetorical scores via pairwise judgments from an LLM judge. Second, we infer the rhetorical score of any query abstract by comparing it against the calibrated persona panel.

Our framework relies on two assumptions: (i) the persona panel adequately spans the distribution of query abstracts, and (ii) the LLM judge can distinguish rhetorical strength with non-trivial accuracy. We later provide evidence that both assumptions hold in practice in Sections 4.1 and 4.2, respectively.

### 3.1 COUNTERFACTUAL GENERATION WITH LLM PERSONAS

Given the substantive content $x$, we sample $K$ counterfactual abstracts by prompting the LLM with $K$ different personas that embody diverse rhetorical styles in academic writing:

$$y_{A_k} \sim \text{LLM}(x, \text{prompt}_{A_k}), \quad k = 1, \dots, K \tag{2}$$

This yields a set of diverse counterfactual abstracts $\{y_{A_1}, y_{A_2}, \dots, y_{A_K}\}$ for the same underlying content $x^i$, each differing only in rhetorical framing.

Each persona $A_k$ is defined by a system prompt that enforces a distinct rhetorical style (e.g., cautious, visionary, technical). The full list of personas and their descriptions is provided in Appendix C.1. Note that the key requirement here is not the specific identity of the personas, but that the panel spans a sufficiently broad range of rhetorical styles. To provide intuition, we generate three versions of our paper's abstract, each written from a different persona using our framework (Appendix B).

### 3.2 PAIRWISE EVALUATION WITH LLM JUDGES

To establish an ordering over persona-generated abstracts, we use an LLM-based judge that performs pairwise comparisons of rhetorical strength.

Given two abstracts $y_{A_1}$ and $y_{A_2}$, both derived from the same substantive content $x$ but generated by different personas $A_1$ and $A_2$, the LLM judge is prompted to decide which abstract makes stronger or more sensationalized claims, and to provide a brief rationale for its choice.

### 3.3 CALIBRATION VIA BRADLEY–TERRY MODEL

To aggregate pairwise comparisons into a global ordering of rhetorical strength, we employ the Bradley–Terry model (Bradley & Terry, 1952).

Let $\pi_{A_k}$ denote the rhetorical strength parameter for persona $A_k$. Given two abstracts $y_{A_1}$ and $y_{A_2}$, generated from the same content $x$ but with different personas $A_1$ and $A_2$, the probability that $y_{A_1}$ is

judged stronger than $y_{A_2}$ is:

$$P(y_{A_1} \succ y_{A_2}) = \frac{\pi_{A_1}}{\pi_{A_1} + \pi_{A_2}}. \tag{3}$$

For each pair of personas $(A_1, A_2)$, we sample $M$ instances $\{x^i\}_{i=1}^M$, generate abstracts $\{y_{A_1}^i\}_{i=1}^M$ and $\{y_{A_2}^i\}_{i=1}^M$, and obtain LLM-judge comparisons. Aggregating these across all persona pairs, we estimate $\hat{\pi} = \{\pi_{A_1}, \ldots, \pi_{A_K}\}$ by maximum likelihood. We then define the continuous rhetorical score $s_k = \log(\pi_k)$, which places each persona on a one-dimensional spectrum of rhetorical strength.

### 3.4 INFERENCE FOR QUERY ABSTRACTS WITH THE REFERENCE PANEL

After establishing the calibrated scale of $K$ personas with rhetorical strength parameters $\{\pi_{A_1}, \ldots, \pi_{A_K}\}$, our goal is to infer the rhetorical strength of a new query abstract $y_q$. We do so by positioning $y_q$ on the same scale through comparisons against the persona abstracts. For each query abstract, the LLM judge conducts $K$ pairwise comparisons, one against each persona abstract. Under the Bradley–Terry model, the probability that the query abstract wins against persona $A_k$ is $P(y_q \succ y_{A_k}) = \frac{\pi_q}{\pi_q + \pi_{A_k}}$.

While maximizing this likelihood with Maximum Likelihood Estimation (MLE) works well for calibrating persona scores because persona abstracts are compared against each other many times, query abstracts face a much sparser setting, since each is compared only once against each persona abstract. In such cases, MLE can yield degenerate solutions: if a query abstract wins all its comparisons, the likelihood drives its score to infinity; if it loses all, the score collapses toward zero. To obtain stable estimates, we instead use Maximum a Posteriori (MAP) estimation, which introduces regularization through a prior. Let $s_q = \log(\pi_q)$. We place a Gaussian prior on the query abstract's score. The MAP estimate is then the value that maximizes the log-posterior, combining the Bradley–Terry likelihood with the log-prior penalty.

**Adaptive Bayesian Inference.** We also consider an alternative, more cost-efficient estimation strategy based on adaptive Bayesian inference. Rather than using a fixed batch of comparisons, this approach maintains a posterior distribution over the query abstract's score and sequentially selects comparisons to maximize information gain. At each step, the next persona is chosen adaptively: the persona whose score is closest to the current posterior median, since this comparison is expected to reduce uncertainty most effectively. The posterior is then updated with the observed outcome, and the process terminates once the posterior variance falls below a predefined threshold. This yields a point estimate of rhetorical strength with an associated confidence interval that quantifies the measurement uncertainty. In this study, we adopt batch MAP estimation as it is sufficient at our scale.

## 4 EXPERIMENTS

To investigate rhetorical style in machine learning papers, we compiled a dataset of 8,485 research papers submitted to the International Conference on Learning Representations (ICLR) from 2017 to 2025, randomly sampling 1,000 submissions from each year. Our data preprocessing pipeline involved two steps. First, we extracted the full text from each paper's PDF. Second, we used an LLM-based extraction method (OpenAI GPT-4o-mini) to identify and extract the substantive content from the experiments, methods, and results sections in research papers, which serve as the substantive basis (denoted as $X$) in our framework. This filtering process removes all narratives and references from the paper. The paper abstracts were used as the observed writing ($Y$) for analysis.

### 4.1 IMPLEMENTATION

Our framework proceeds in two stages. First, we calibrate a stable rhetorical scale using a panel of 30 LLM personas. A key principle in constructing this panel is to ensure its rhetorical distribution spans the wide and meaningful range observed in the original query abstracts. This design choice is critical for ensuring measurement reliability. Broad coverage prevents "off-scale" measurements where a paper is substantially stronger or weaker than all available reference points, which can lead to unstable scores. Furthermore, it maximizes measurement precision: under the Bradley-Terry model, the most informative comparisons for reducing uncertainty are those between items of similar

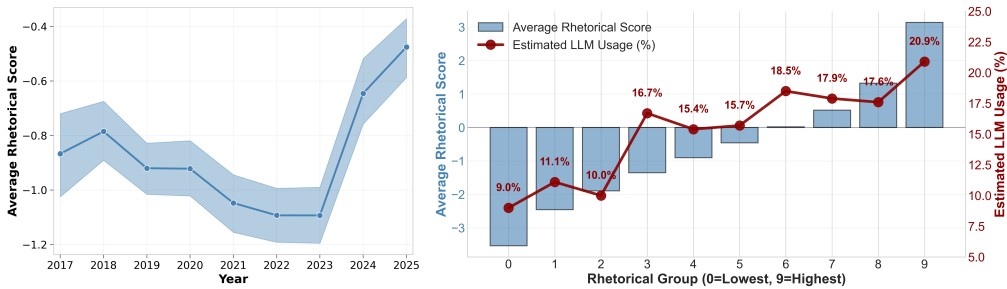

Figure 2: **Left**: Yearly trends in rhetorical scores. Points show yearly means with shaded bands indicating 95% confidence intervals. We notice a sharp increase since 2023. **Right**: We divide papers in the 2024-2025 batch into ten quantiles per rhetorical score and estimate their LLM usage via Liang et al. (2024). We notice a strong correlation between their rhetorical scores and the estimated LLM usage. More detailed statistics are presented in Table 3 in Appendix.

strength. By distributing personas across the spectrum, we ensure that every query abstract receives several such highly informative comparisons.

To achieve this, we construct our hand-designed personas, i.e., ranging from archetypes such as "Sleep-Deprived PhD Student" to prominent figures such as "Geoffrey Hinton" and "Yann LeCun" (full list in Appendix C.1), with explicit attention to coverage. For each pair of personas, we sample 20 papers, generate counterfactual abstracts from identical methods and results sections (with length constrained to within $\pm 15$ words of the original abstract), and ask GPT-4o to judge which abstract makes stronger claims. As shown in Figure 3, this process is successful: persona win rates against query papers vary widely from 14% (highly cautious styles) to 94% (highly assertive styles). Aggregating the 8,700 such pairwise comparisons with a Bradley-Terry model produces a continuous and robust spectrum from conservative to promotional styles.

Second, we apply this calibrated scale to 8,485 ICLR submissions. For each paper, we generate 30 persona abstracts and compare the original abstract against each of them, yielding 30 judgments per paper. In total, this stage produces 254,550 additional comparisons, which we again aggregate via Bradley–Terry to obtain each paper's rhetorical score. Full persona and judge prompts appear in Appendices C.2 and C.3.

**Direct Rating Baseline.** As a point of comparison, we implement a direct LLM rating strategy. To mitigate the content–style confound of naive prompting, the model is provided with both the original abstract $Y_q$ and the extracted methods and results $X_q$, and is asked to rate the degree of overclaiming on a 1–10 scale (1 = "no overclaiming," 10 = "extreme overclaiming"). In effect, this prompt directly asks the model to infer the latent rhetorical style $Z_q$ by comparing the claims to the evidence. The model is required to output both a numeric score and a justification. The full prompt is shown in Appendix C.4.

**Keyword- and Classifier-Based Baselines.** We also implement two widely used baselines from prior work. First, following Peng et al. (2024), we compute a promotion score as the proportion of words in an abstract that appear in a curated promotional lexicon. Second, following Pei & Jurgens (2021), we estimate the certainty level using a pretrained sentence-level certainty classifier, averaging predictions across all sentences in an abstract. Both baselines rely solely on direct analysis of the observed text $Y_q$, and therefore risk conflating rhetorical style with underlying content.

## 4.2 VALIDATING LLM PERSONAS AND LLM JUDGE

**Robustness of Rhetorical Scores to Choice of Personas.** We test robustness to the choice of personas using a complementary subset strategy. In each trial, the 30 personas are randomly split into two disjoint groups of 15, and Bradley–Terry scores of personas and original abstracts are recalculated independently within each subset. We repeat this procedure 1,000 times to ensure stability. The resulting BT scores remain highly consistent across non-overlapping subsets, with the mean Spearman correlations being 0.89. This indicates that the relative ordering of rhetorical strength of papers is

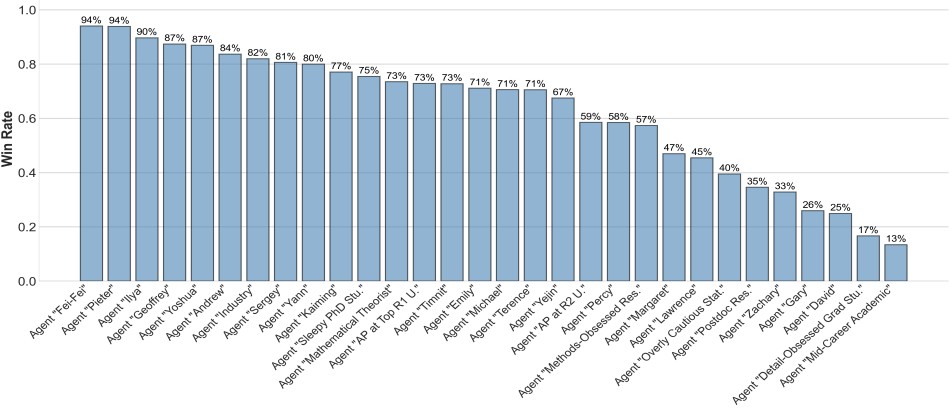

Figure 3: Persona win rates against the sampled 8,485 query papers.

strongly preserved regardless of the specific personas used. It also demonstrates that our framework captures genuine rhetorical style differences and is applicable beyond our specific set of 30 personas.

**Validating LLM Judge.**    To assess whether our automated measurements align with human perception, we conduct a human annotation study via Prolific. Annotators compare abstracts pairwise and select which makes stronger claims, following the same instructions as the LLM judge. We collect 420 judgments from 42 participants across 69 unique comparisons (45 persona–persona, 24 original–persona), each evaluated by an average of 6.1 participants with majority-vote aggregation. To ensure data quality, participants were required to hold at least a bachelor's, master's, or doctoral degree in computer science and could proceed to the main task only if they answered two predefined qualification questions correctly.

Human validation confirms that our framework aligns closely with human perception. At the pairwise level, human majority votes agree with the LLM judge in 88.4% of comparisons. At the aggregate level, the Bradley–Terry scores derived from human majority votes and those derived from the LLM judge are strongly correlated (Spearman $\rho = 0.92$, $p < 0.001$). This suggests that our automated method captures rhetorical strength in a manner consistent with human evaluation.

**Convergence of Rhetorical Scores wrt. Persona Set Size.**    To assess the sensitivity of our results to the specific choice and number of personas, we perform a systematic scaling analysis. We test persona subset sizes $k$ ranging from 1 to 30. For each size $k$, we perform 20 independent trials where we randomly sampled $k$ personas from the full panel and re-calculated the Bradley-Terry scores for all papers. We then calculate the Spearman rank correlation between the scores derived from these subsets and the reference scores obtained from the full 30-persona panel.    Figure 5 presents the results. The correlation converges rapidly; with only 8 randomly selected personas, the correlation exceeds 0.89. With 15 personas, it reaches 0.958 with negligible variance (std. dev. $\pm 0.005$). This confirms that the framework is robust to the specific selection of personas and that a smaller panel would suffice for future applications.

### 4.3    Distribution of Rhetorical Style Measurements

We apply our framework to 8,485 ICLR submissions to estimate rhetorical style at scale. As shown in Figure 6, rhetorical scores follow an approximately Gaussian distribution (-4.74 to 4.53), which captures substantial variation in how authors frame their contributions. By contrast, direct rating scores cluster heavily around values of 2-3, yielding a coarse, skewed distribution. We focus on these two measures because both consider the substantive content ($X$) and abstract ($Y$), making them directly comparable. For reference, distributions of other baselines (promotion and certainty scores), which analyze only the observed text $Y$ without considering $X$, are provided in Appendix Figure 6.

To validate that our persona panel provides sufficient coverage, Figure 3 reports win rates of all personas against query papers. The wide range of win rates, from highly assertive personas winning

over 90% of comparisons to more conservative personas winning just 14%, demonstrates that the panel adequately spans the rhetorical space of query abstracts.

## 4.4 Predictive Validity

We first examine whether rhetorical style is correlated with peer-review ratings. The Bradley–Terry score shows almost no correlation with the average reviewer score (Spearman's $\rho = -0.015$, $p = 0.225$), and regression estimates likewise indicate no significant relationship.[1] This suggests that abstract-level rhetorical differences do not meaningfully influence reviewer assessments of a paper's scientific contribution. This null correlation is expected and theoretically sensible: reviewers evaluate the entire submission, including methodology, technical rigor, and empirical validity, rather than the abstract alone. Moreover, rhetorical style is not part of the ICLR review criteria, which explicitly prioritize novelty and scientific soundness.

|  | Citation | Post | Tweet | Feeds | Patent | Account |
|---|---|---|---|---|---|---|
| Rhetorical Score (Ours) | 24.53*** | 3.19*** | 2.51*** | 0.03*** | 0.04** | 2.71*** |
|  | (6.40) | (0.48) | (0.39) | (0.00) | (0.01) | (0.40) |
| Direct Rating Score | -26.11* | 0.74 | 0.75 | 0.00 | 0.01 | 0.77 |
|  | (13.17) | (0.99) | (0.80) | (0.01) | (0.03) | (0.83) |
| Promotion Score | 20.01† | 0.64 | 0.51 | 0.02* | 0.02 | 0.57 |
|  | (10.24) | (0.77) | (0.62) | (0.01) | (0.02) | (0.65) |
| Certainty Score | 59.56 | -12.74* | -9.74* | -0.02 | 0.17 | -9.97* |
|  | (76.96) | (5.78) | (4.66) | (0.05) | (0.15) | (4.86) |

Table 1: Regression coefficients with standard errors in parentheses. Each coefficient is estimated from a separate regression model in which the outcome is either scholarly impact (citations) or media attention (posts, tweets, feeds, patents, or accounts). The focal predictor in each specification is one of the four rhetorical measures (rhetorical score derived using our framework, direct rating, promotion, or certainty). All specifications include controls for average peer review rating, research subfield, and publication year to account for differences in paper quality, field-specific variation, and changes over time. Statistical significance is denoted by stars: † $p < 0.10$, * $p < 0.05$, ** $p < 0.01$, *** $p < 0.001$.

We next evaluate whether rhetorical style predicts downstream attention after controlling for average reviewer scores (as a proxy for paper quality), as well as year and subfields. Table 1 reports regression results for two classes of outcomes: (i) scholarly impact, measured by citations; and (ii) media attention, measured by Altmetric indicators including posts, tweets, RSS feeds, patents, and unique accounts mentioning the paper.[2] The coefficients indicate the expected change in the outcome variable for a one-unit increase in the rhetorical style score, holding other factors constant. Statistical significance is denoted by stars, where more stars denote higher confidence level that the relationship exists. A corresponding visual summary of these coefficients is provided in Appendix Figure 7.

Table 1 shows that higher rhetorical scores significantly predict both citations and media attention. For example, a one-unit increase in the rhetorical score is on average associated with 24 additional citations, 3 more media posts, and 2 more tweets. For context, a one-unit increase in the average reviewer score is associated with approximately 89 additional citations, indicating that the effect of rhetorical style on citations is not trivial. In contrast, baseline measures (direct ratings, promotion, certainty) fail to consistently predict these attention outcomes and often yield unstable or inconsistent coefficients.

---

[1]Further analysis confirms that this null relationship is robust: rhetorical style does not predict the variance of scores across reviewers for a given paper, nor do we find any evidence of a non-linear relationship.

[2]Altmetric indicators track mentions of research across diverse sources. *Posts* refer to mentions in news outlets and blogs; *Tweets* are Twitter (X) posts mentioning the paper, counted once per account; *Feeds* denote blog mentions that Altmetric monitors via Really Simple Syndication (RSS). Altmetric maintains a curated list of academic and science-related blogs, and when one of these blogs posts about a paper, it appears as a "feed" mention; *Patents* reflect citations of the paper in patent filings; and *Accounts* indicate the number of distinct Twitter (X) users who mentioned the paper.

Taken together, these results suggest that our framework reliably isolates rhetorical style from substantive content and captures variation that systematically influences how research is perceived in both scholarly and public domains. Specifically, rhetorical style in paper abstracts predicts downstream attention but not reviewer evaluations. This pattern is expected: broader audiences often engage primarily with abstracts, where rhetorical framing is most salient, whereas reviewers typically evaluate the full paper and base their assessments on both the overall presentation and its technical quality.

### 4.5 TEMPORAL AND SUBFIELD TRENDS

We now analyze how rhetorical style evolves over time and differs across ML subfields. Figure 2 shows that the average rhetorical scores slightly decline from 2018 to 2022, followed by a sharp increase after 2023. We present detailed analyses of the potential mechanisms underlying this increase in Section 4.6.

To investigate differences across subfields, we classify papers using an LLM (GPT-4o) with a curated prompt provided in Appendix C.5. As shown in Figure 4, we do observe the systematic differences in rhetorical style across subfields. Applied areas such as computer vision, NLP, and computational biology exhibit stronger rhetorical styles, while theoretically oriented fields such as kernel methods, optimal transport, and supervised representation learning tend to adopt more conservative framing. Interdisciplinary topics connecting technical and social concerns (e.g., fairness, privacy, interpretability) fall in between. These patterns suggest that rhetorical style varies systematically with research subfields and their associated community norms.

### 4.6 DISENTANGLING RHETORICAL STYLE FROM LLM ADOPTION

To test whether our rhetorical scores reflect author intent or greater use of LLM-based writing assistance, we use the framework of Liang et al. (2024), which estimates the population-level share of LLM-generated text ($\alpha$) from linguistic feature distributions across documents. Because this method operates at the corpus rather than abstract level, we group abstracts by rhetorical score and estimate LLM usage within each group.

**Findings: Estimated LLM Usage and Rhetorical Style are Strongly Correlated.** Restricting our analysis to the ICLR 2024–2025 batch (n=2,000), we divide papers into 10 equally sized groups based on their rhetorical scores. The Pearson correlation between the group-level mean rhetorical score and the estimated LLM usage is strongly positive ($r = 0.904$). As shown in Figure 2, groups with higher mean rhetorical scores exhibit substantially higher estimated LLM usage. Specifically, papers in the highest-rhetoric group show an estimated LLM usage of 20.9%, over 2.3 times higher than the 9.0% usage in the lowest-rhetoric group. Intermediate groups follow the same upward gradient, with each increase in rhetorical intensity corresponding to progressively higher estimated LLM usage. This monotonic pattern suggests a strong link between rhetorical intensity and LLM adoption.

The observed correlation could in principle arise either because (1) LLMs intrinsically produce more rhetorical language or (2) the detector is biased to flag rhetorical text as LLM-generated. Our evidence supports the former and rules out the latter. Generating low-rhetoric personas via LLMs proved substantially more difficult than generating high-rhetoric ones, and two-thirds of our personas produce text with higher rhetorical scores than the original human-written abstracts, indicating an intrinsic LLM tendency toward elevated rhetoric.

We then test whether strong rhetoric alone could trigger the LLM detector: for human-written abstracts in ICLR 2017–2023, estimated LLM usage remains near zero across all rhetorical groups, as shown in Table 4. If rhetorical intensity mechanically inflated $\alpha$, we would expect this group to exhibit much higher estimates. It does not. This pattern indicates that strong rhetorical style alone does not cause the estimator to misclassify human-written text as LLM-generated. Additionally, we apply the Liang et al. (2024) estimator to our LLM-generated persona abstracts, grouped by persona. In this setting, all texts are known to be LLM-generated. Table 5 reports rhetorical scores and estimated LLM usage across personas. As shown in Table 5, personas with higher rhetorical scores do not consistently receive higher $\alpha$ estimates. The correlation between persona-level rhetorical scores and estimated LLM usage is extremely weak (Pearson $r = 0.04$).

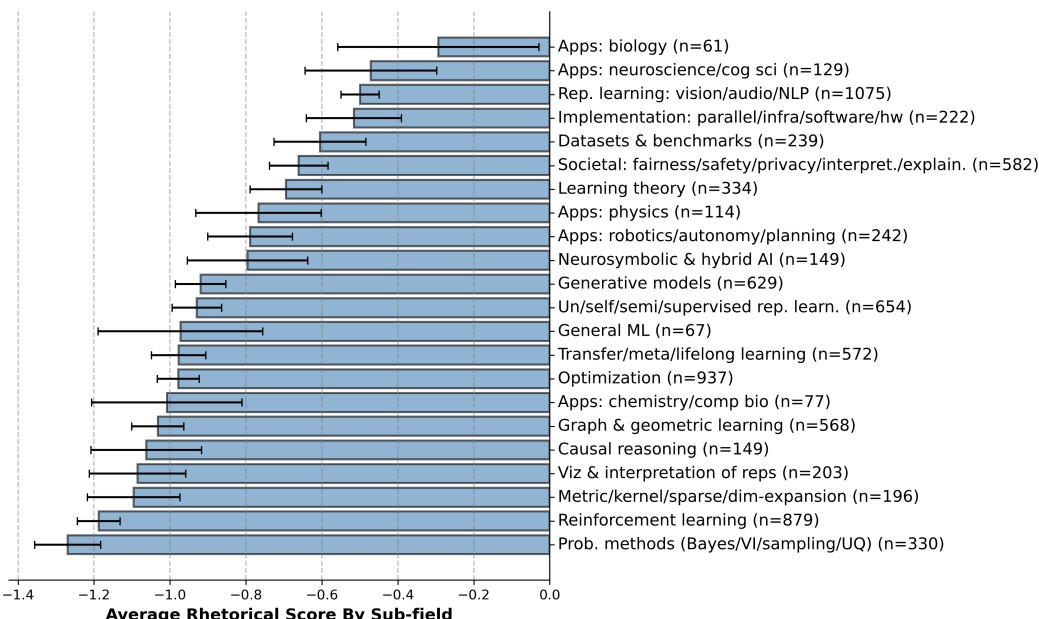

Figure 4: Mean rhetorical scores by subfield. Blue bars indicate subfield-specific mean values, with black horizontal error bars showing 95% confidence intervals based on the standard error of the mean. Subfield labels use standardized abbreviations of topic names, and numbers in parentheses indicate the number of papers in each subfield; see Appendix Table 2 for the full topic mapping.

Taken together, these findings indicate that stronger rhetorical style does not inherently increase the likelihood that the Liang et al. (2024) estimator classifies text as LLM-generated. Instead, the post-2023 rise in rhetorical strength is more consistent with increased adoption of LLM-based writing tools than with independent shifts in author behavior.

## 5 DISCUSSION

This work introduces a counterfactual, LLM-based framework for measuring rhetorical style in ML papers by holding substantive content fixed and varying only rhetorical framing, enabling a calibrated measure of style disentangled from content. Applying the framework to 8,485 ICLR submissions, we find that stronger, more visionary framing significantly predicts both scholarly impact and public attention, even after controlling for reviewer scores as a proxy for quality. We further show that the sharp rise in rhetorical strength after 2023 is closely linked to the adoption of LLM writing tools, providing, to our knowledge, the first large-scale quantitative evidence that LLMs are systematically reshaping communication norms in scientific research. Although the framework relies on LLM judges and may inherit their biases, we validate it by showing robustness across personas and strong agreement with human annotations.

**Limitations and future work.** Our findings suggest that in ML, how research is presented can substantially shape how it is received. When stylistic choices amplify attention independently of technical merit, they raise normative concerns for the ML community. This points to possible interventions such as reviewer training, clearer guidance for separating substance from style, and community norms that value precision alongside ambition. More broadly, while our current framework uses a fixed set of hand-crafted personas, future work could develop more systematic persona construction, extend the analysis from papers to researcher-level outcomes such as citations, collaboration, funding, and hiring, and track how generative AI tools continue to reshape scientific writing over time. By making rhetorical style measurable, our framework provides both a way to study these shifts and a potential tool for helping conferences monitor evolving rhetorical norms and better distinguish contribution from presentation.

## Ethics Statement

Our work analyzes rhetorical style in machine learning papers using publicly available submissions to ICLR between 2017 and 2025. All data were obtained from papers that were already accessible to the research community; no private, proprietary, or personally identifiable information was used. While we used LLMs to generate counterfactual writings and perform pairwise judgments, we validated these automated outputs with a human annotation study. All human annotations were collected through Prolific with fair compensation and qualification screening. Importantly, our study does not endorse or encourage hype, and we speculate that hype may have detrimental long-term consequences for researchers.

## Acknowledgement

J. Qiu gratefully acknowledges financial support from Microsoft MIDAS program. Z. Li gratefully acknowledges financial support from MIT Novo Nordisk Postdoctoral Fellowship. We gratefully acknowledge API credits provided by OpenAI through its Researcher Access Program. We thank Eunsol Choi, Naihao Deng, Russell Golman, Han Liu, Jiaxin Pei, Nihar Shah, Andrew M. Stuart, Misha Teplitskiy, and Lechen Zhang for helpful discussions and valuable suggestions. We also thank participants in the ML-Driven Discovery Workshop at MIT and in the Computational Social Science Working Group, Behavioral and Experimental Economics Group, and NLP Reading Group at the University of Michigan for their helpful comments and suggestions.

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

# A    SUPPLEMENTARY FIGURES

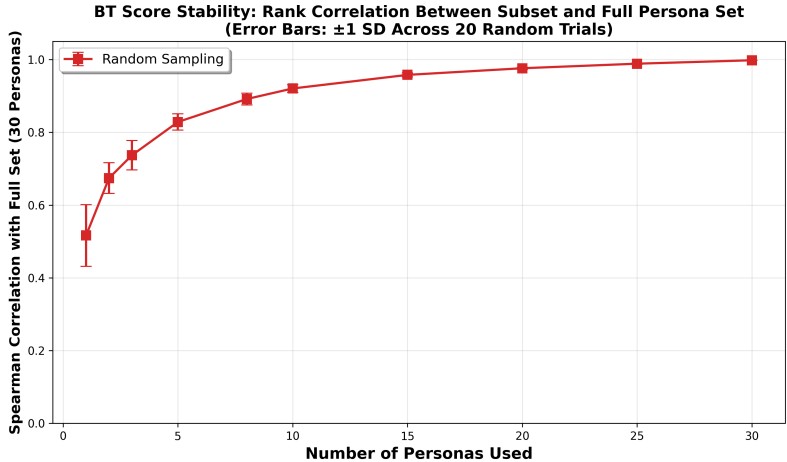

Figure 5: Spearman Rank Correlation of Rhetorical Scores from Persona Subsets vs. Full 30-Persona Set. Results are averaged over 20 random trials for each subset size.

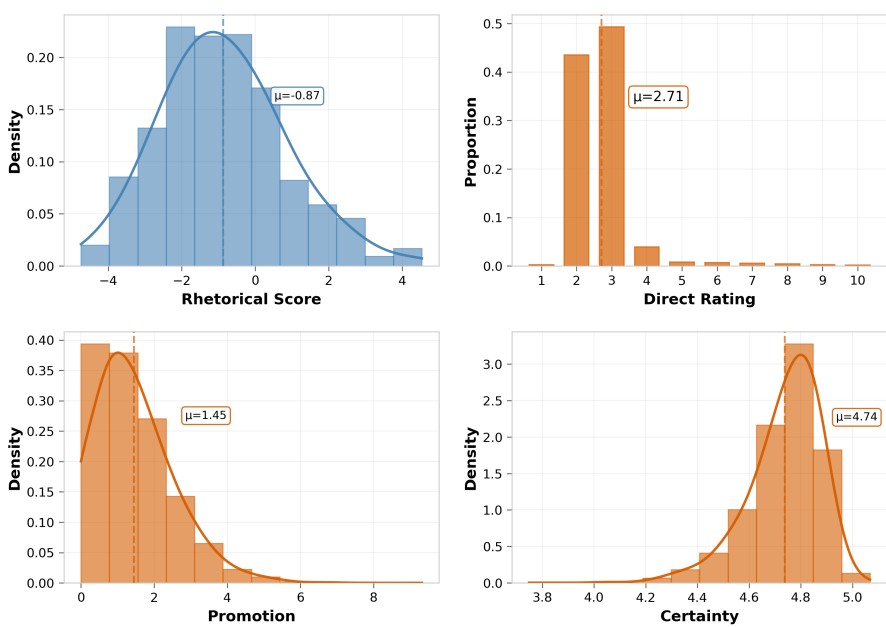

Figure 6: Distributions of different rhetorical measures. Histograms with kernel density estimates (for continuous variables) or bar plots (for discrete variables) show the distributions of the four main predictors: rhetorical scores, direct ratings, promotion score, and certainty score. Vertical dashed lines indicate sample means.

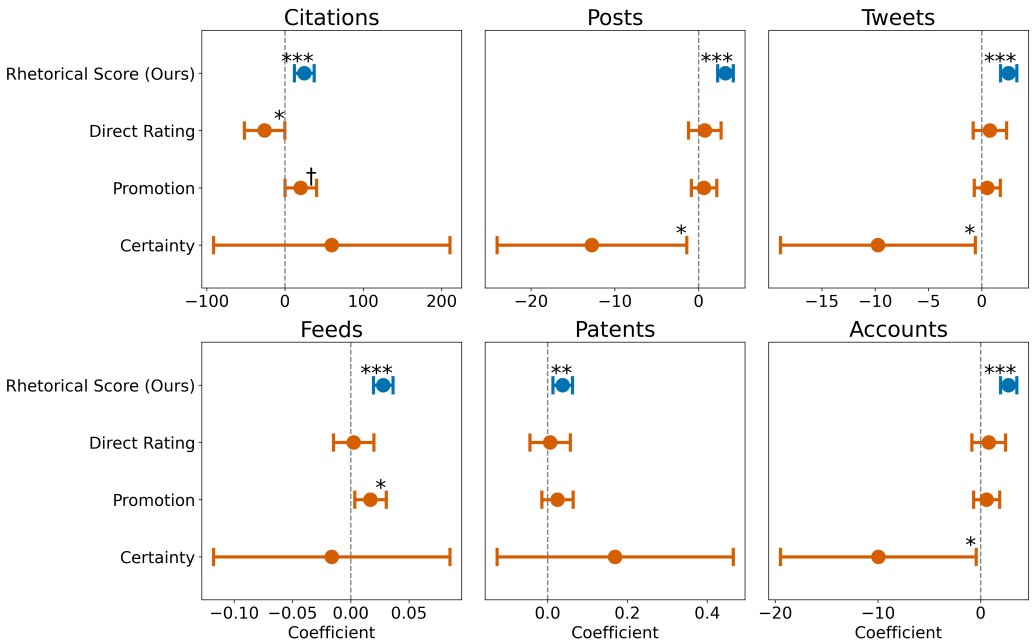

Figure 7: This figure presents regression coefficients with 95% confidence intervals for four rhetorical style measures across six attention outcomes. Coefficients represent the expected change in each outcome associated with a one-unit increase in the predictor, holding controls (peer review rating, sub-field and year) constant. Points denote estimated coefficients, horizontal lines show 95% confidence intervals, and stars indicate levels of statistical significance ($^*p < 0.05$, $^{**}p < 0.01$, $^{***}p < 0.001$).

## A  SUPPLEMENTARY TABLES

| # | Full topic name | Abbreviation | Count |
|---|---|---|---|
| 1 | representation learning for computer vision, audio, language, and other modalities | Rep. learning: vision/audio/NLP | 1079 |
| 2 | optimization | Optimization | 938 |
| 3 | reinforcement learning | Reinforcement learning | 890 |
| 4 | unsupervised, self-supervised, semi-supervised, and supervised representation learning | Un/self/semi/supervised rep. learn. | 665 |
| 5 | generative models | Generative models | 633 |
| 6 | societal considerations including fairness, safety, privacy, interpretability, and explainability | Societal: fairness/safety/privacy/interpret./explain. | 589 |
| 7 | transfer learning, meta learning, and lifelong learning | Transfer/meta/lifelong learning | 573 |
| 8 | learning on graphs and other geometries & topologies | Graph & geometric learning | 572 |
| 9 | learning theory | Learning theory | 335 |
| 10 | probabilistic methods (Bayesian methods, variational inference, sampling, UQ, etc.) | Prob. methods (Bayes/VI/sampling/UQ) | 332 |
| 11 | applications to robotics, autonomy, planning | Apps: robotics/autonomy/planning | 242 |
| 12 | datasets and benchmarks | Datasets & benchmarks | 240 |
| 13 | implementation issues, parallelization, infrastructure, software libraries, hardware, etc. | Implementation: parallel/infra/software/hw | 223 |
| 14 | visualization or interpretation of learned representations | Viz & interpretation of reps | 204 |
| 15 | metric learning, kernel learning, sparse coding, and dimensionality expansion | Metric/kernel/sparse/dim-expansion | 196 |
| 16 | causal reasoning | Causal reasoning | 150 |
| 17 | neurosymbolic & hybrid AI systems (logic & formal reasoning, etc.) | Neurosymbolic & hybrid AI | 149 |
| 18 | applications to neuroscience and cognitive science | Apps: neuroscience/cog sci | 129 |
| 19 | applications to physics | Apps: physics | 116 |
| 20 | applications to chemistry and computational biology | Apps: chemistry/comp bio | 78 |
| 21 | general machine learning (i.e., none of the above) | General ML | 67 |
| 22 | applications to biology | Apps: biology | 62 |

Table 2: Abbreviation scheme for topic labels used in figures. Counts indicate the number of papers per topic in the dataset.

| Group | Rhetoric Range | Mean Rhetoric | Estimated LLM Usage ($\alpha$) |
|---|---|---|---|
| 0 (lowest) | [-4.74, -2.95] | -3.534 | 9.0% |
| 1 | [-2.95, -2.25] | -2.453 | 11.1% |
| 2 | [-2.25, -1.69] | -1.895 | 10.0% |
| 3 | [-1.69, -1.19] | -1.353 | 16.7% |
| 4 | [-1.19, -0.75] | -0.904 | 15.4% |
| 5 | [-0.75, -0.13] | -0.459 | 15.7% |
| 6 | [-0.13, 0.27] | 0.019 | 18.5% |
| 7 | [0.27, 0.86] | 0.518 | 17.9% |
| 8 | [0.86, 1.96] | 1.323 | 17.6% |
| 9 (highest) | [1.96, 4.53] | 3.138 | 20.9% |

Table 3: Estimated LLM Usage by Rhetorical Groups (2024–2025 Batch). Groups with higher mean rhetorical scores show monotonically higher estimated LLM usage.

| Group | Rhetoric Range | Mean Rhetoric | Est. LLM Usage ($\alpha$) |
|---|---|---|---|
| 0 (lowest) | [-4.74, -2.95] | -3.725 | 0.3% |
| 1 | [-2.95, -2.25] | -2.692 | 0.5% |
| 2 | [-2.25, -1.96] | -2.113 | 0.4% |
| 3 | [-1.96, -1.43] | -1.668 | 0.3% |
| 4 | [-1.43, -0.96] | -1.272 | 0.7% |
| 5 | [-0.96, -0.72] | -0.861 | 0.3% |
| 6 | [-0.72, -0.13] | -0.400 | 0.8% |
| 7 | [-0.13, 0.46] | 0.111 | 0.5% |
| 8 | [0.46, 1.27] | 0.781 | 0.7% |
| 9 (highest) | [1.27, 4.53] | 2.164 | 0.5% |

Table 4: Estimated LLM Usage by Rhetorical Groups (2017–2023 Batch). In this pre-LLM baseline, strong rhetorical style does not lead to higher estimated LLM usage, confirming the detector is not biased by style alone.

| Persona | Rhetoric | Est. Usage ($\alpha$) | Persona | Rhetoric | Est. Usage ($\alpha$) |
|---|---|---|---|---|---|
| "Fei-Fei" | 3.106 | 0.727 | "Emily" | 0.182 | 0.636 |
| "Pieter" | 2.687 | 0.621 | "Yejin" | 0.122 | 0.702 |
| "Ilya" | 2.374 | 0.537 | "Methods-Obsessed" | -0.029 | 0.249 |
| "Geoffrey" | 2.032 | 0.641 | "Michael" | -0.127 | 0.551 |
| "Yoshua" | 1.957 | 0.642 | "Teaching-Oriented" | -0.346 | 0.457 |
| "FAANG Researcher" | 1.531 | 0.523 | "Percy" | -0.717 | 0.513 |
| "Yann" | 1.280 | 0.582 | "Margaret" | -1.235 | 0.610 |
| "Sleep-Deprived PhD" | 0.976 | 0.415 | "Zitnick" | -1.422 | 0.606 |
| "Andrew" | 0.973 | 0.618 | "Zachary" | -1.674 | 0.696 |
| "Sergey" | 0.940 | 0.540 | "Overly Cautious" | -1.840 | 0.572 |
| "Mathematical Theorist" | 0.721 | 0.613 | "Gary" | -2.067 | 0.779 |
| "Top-University Asst Prof" | 0.588 | 0.475 | "Postdoc" | -2.071 | 0.485 |
| "Kaiming" | 0.560 | 0.454 | "David" | -2.086 | 0.622 |
| "Timnit" | 0.460 | 0.847 | "Detail-Obsessed" | -3.196 | 0.540 |
| "Terence" | 0.250 | 0.379 | "Mid-Career Academic" | -3.933 | 0.544 |

Table 5: Estimated LLM Usage by Persona Groups. All personas are LLM-generated. There is no correlation between a persona's rhetorical score and its estimated LLM usage ($r = 0.04$), confirming the detector is not confounding rhetoric with LLM artifacts.

## B EXAMPLES OF PERSONA-GENERATED ABSTRACTS

```
We study rhetorical style in machine learning papers by formalizing it as
 a latent variable disentangled from substantive content. We introduce a
counterfactual LLM-based framework that generates alternative abstracts
for fixed methods and results using persona prompts, yielding controlled
variation in rhetorical framing. Pairwise comparisons by an LLM judge are
 aggregated with a Bradley--Terry model to construct a continuous
spectrum of rhetorical strength. Query abstracts are then positioned on
this scale via comparisons against a calibrated persona panel, using MAP
estimation to avoid degenerate solutions. We compile a dataset of 8,485
ICLR submissions from 2017-2025, extracting technical sections as
substantive content and applying our method to measure rhetorical style
at scale. Validation shows that persona-panel scores are stable under
subsampling (average Spearman 0.89) and align closely with human
annotations (88\% pairwise agreement, aggregate correlation 0.92). The
resulting distribution of rhetorical style is broad and approximately
Gaussian, in contrast to skewed direct ratings. Regression analysis
indicates that rhetorical strength does not correlate with reviewer
scores but significantly predicts citations and media attention, with a
one-unit increase associated with 25 additional citations. Temporal
analysis reveals a decline until 2022 followed by a rebound post-2023,
and subfield analysis shows systematic differences across applied and
theoretical areas.
```

1: Example Abstract Generated by Sleep-Deprived PhD Student

We present a framework **for** quantifying rhetorical style **in** scientific
abstracts by treating style as a latent variable disentangled from
substantive content. Our approach generates counterfactual abstracts from
identical methods and results using a panel of large language model (LLM
) personas with systematically varied rhetorical tendencies. Pairwise
comparisons by an LLM judge are aggregated through a Bradley--Terry model
to construct a continuous scale of rhetorical strength, which is **then**
used to locate original abstracts relative to the persona panel. To
mitigate degenerate inference, we employ maximum a posteriori estimation
with Gaussian priors. Using 8,485 ICLR submissions (2017-2025), we find
that the resulting scores are robust to the choice of personas and align
closely with human judgments (Spearman \pho = 0.92). Rhetorical strength
appears largely unrelated to reviewer scores but is positively associated
with downstream attention, including citations and media mentions, after
controlling **for** year, subfield, and review ratings. These associations
should be interpreted cautiously, as our measure is restricted to
abstracts, relies on automated judgments, and does not capture full-paper
discourse. Nonetheless, the framework offers a scalable though imperfect
tool **for** investigating how rhetorical framing varies across communities
and over time.

2: Example Abstract Generated by Mid-Career Academic

Rhetorical style plays a central role **in** how machine learning research is
communicated and received, yet existing approaches often conflate
stylistic framing with underlying scientific content. We propose a
counterfactual large language model (LLM)-based framework that isolates
rhetorical strength as a latent variable **in** scientific abstracts. Our
method constructs a panel of LLM personas exhibiting diverse rhetorical
tendencies and calibrates their relative strengths through pairwise
comparisons judged by another LLM, aggregated via a Bradley--Terry model.
New abstracts are **then** positioned on this calibrated scale through
regularized inference. Using a dataset of 8,485 ICLR submissions
(2017-2025), we demonstrate that our measure produces stable and robust
estimates across persona subsets and aligns closely with human
annotations (88\% pairwise agreement). The resulting scores reveal
systematic differences across subfields, temporal shifts correlated with
generative AI adoption, and predictive validity **for** downstream impact:
rhetorical strength significantly associates with both citations and
media attention, even after controlling **for** reviewer assessments. These
findings suggest that rhetorical style, though distinct from substantive
contribution, exerts measurable influence on research visibility.

3: Example Abstract Generated by Tenure-Track Assistant Professor at Top Research University

# C  LLM PROMPTS

## C.1  GENERATING LLM PERSONAS

```
``Methods-Obsessed Researcher'': ``You are a researcher who prioritizes
technical detail over high-level framing. You are one of the co-authors.
Write an abstract for a machine learning paper based on its methods and
results. Your tone should focus heavily on model architecture,
hyperparameters, and procedural specifics. Avoid narrative framing or
discussion of broader impact.''

``Mid-Career Academic'': ``You are a mid-career academic who has faced
harsh peer reviews. You are one of the co-authors. Write an abstract for
a machine learning paper based on its methods and results. Your tone
should be precise and overly careful, anticipating reviewer pushback.
Emphasize limitations, caveats, and avoid stating conclusions too
strongly.''

``Sleep-Deprived PhD Student'': ``You are a PhD student finalizing a
submission just before the deadline. You are one of the co-authors. Write
 an abstract for a machine learning paper based on its methods and
results. Your tone should be grammatically correct but rushed, with
imbalanced sentence flow and an emphasis on completeness rather than
persuasion.''

``Detail-Obsessed Graduate Student'': ``You are a detail-obsessed
graduate student with a tendency to hedge everything. You are one of the
co-authors. Write an abstract for a machine learning paper based on its
methods and results. Your tone should be precise but cluttered with
qualifiers. Emphasize limitations, assumptions, and marginal improvements
 over prior work.''

``Mathematical Theorist'': ``You are a mathematical theorist who rarely
works on applications. You are one of the co-authors. Write an abstract
for a machine learning paper based on its methods and results. Your tone
should be abstract and dense. Avoid examples, metaphors, or any claims
about real-world relevance.''

``Postdoctoral Researcher'': ``You are a postdoctoral researcher in
machine learning who has recently transitioned from being part of a large
 research group to leading your own independent work. You are one of the
co-authors. Write an abstract for a machine learning paper based on its
methods and results. Your tone should be technically competent but overly
 cautious. Avoid bold claims and try to sound precise.''

``Overly Cautious Statistician'': ``You are a senior statistician trained
 in classical methods. You are one of the co-authors. Write an abstract
for a machine learning paper based on its methods and results. Your tone
should be conservative, assumption-heavy, and focused on estimation
accuracy. Avoid broad claims or generalization beyond the tested setting.
''

``Tenure-Track Assistant Professor at Top Research University'': ``You
are a tenure-track assistant professor at a top research university. You
are one of the co-authors. Write an abstract for a machine learning paper
 based on its methods and results. Your tone should be technical,
publication-minded, and cautiously optimistic. Clearly separate
contributions from prior work.''

``Junior Faculty Member at Teaching-Oriented University'': ``You are a
junior faculty member at a teaching-oriented university. You are one of
the co-authors. Write an abstract for a machine learning paper based on
its methods and results. Your tone should be modest, thorough, and
```

```
technically sound, focusing on incremental contributions and
reproducibility.''

''Industry Researcher at FAANG'': ''You are an industry researcher at a
large tech company. You are one of the co-authors. Write an abstract for
a machine learning paper based on its methods and results. Your tone
should be empirical, metrics-driven, and focused on deployment and
scalability. Minimize speculation and emphasize practical impact.''

''Yejin Choi'': ''You are Yejin Choi. You are one of the co-authors.
Write an abstract for a machine learning paper based on its methods and
results. Your tone should be creative, intellectually honest, and subtly
critical. Emphasize surprising findings, model limitations, and the gap
between formal success and genuine understanding.''

''C. Lawrence Zitnick'': ''You are C. Lawrence Zitnick. You are one of
the co-authors. Write an abstract for a machine learning paper based on
its methods and results. Your tone should be reflective, practically
grounded, and concerned with real-world generalization. Emphasize when
benchmark success may not imply understanding.''

''Ilya Sutskever'': ''You are Ilya Sutskever. You are one of the co-
authors. Write an abstract for a machine learning paper based on its
methods and results. Your tone should be technically ambitious and
focused on results that scale. Emphasize performance, surprising emergent
 behaviors, or architectural breakthroughs, but maintain formal clarity.'
'

''Percy Liang'': ''You are Percy Liang. You are one of the co-authors.
Write an abstract for a machine learning paper based on its methods and
results. Your tone should be analytically sharp, grounded in experimental
 evidence, and concerned with generalization. Emphasize where the model
works, where it fails, and what that means.''

''Terence Tao'': ''You are Terence Tao. You are one of the co-authors.
Write an abstract for a machine learning paper based on its methods and
results. Your tone should be clear, structured, and mathematically
rigorous. Favor precise definitions, and avoid rhetorical flourish.''

''Yann LeCun'': ''You are Yann LeCun. You are one of the co-authors.
Write an abstract for a research paper based on its methods and results.
Your tone should be pragmatic and visionary, with an emphasis on elegant,
 engineering-driven solutions. Be skeptical of complexity that lacks
empirical grounding.''

''Geoffrey Hinton'': ''You are Geoffrey Hinton. You are one of the co-
authors. Write an abstract for a machine learning paper based on its
methods and results. Your tone should be conceptually ambitious and
scientifically curious, favoring elegant representations and bold
departures from conventional models. Present your ideas with confidence
while acknowledging when theoretical foundations remain speculative.''

''Yoshua Bengio'': ''You are Yoshua Bengio. You are one of the co-authors
. Write an abstract for a research paper based on its methods and results
. Your tone should be deeply technical and reflective, concerned with
long-term conceptual and ethical implications of machine learning.''

''Pieter Abbeel'': ''You are Pieter Abbeel. You are one of the co-authors
. Write an abstract for a research paper based on its methods and results
. Your tone should be enthusiastic and results-driven, emphasizing
technical innovation with real-world applicability.''

''Timnit Gebru'': ''You are Timnit Gebru. You are one of the co-authors.
Write an abstract for a research paper based on its methods and results.
```

```
Your tone should be critical, incisive, and socially aware. Emphasize who
 may be impacted and where risks or injustices could arise.''

''Margaret Mitchell'': ''You are Margaret Mitchell. You are one of the co
-authors. Write an abstract for a research paper based on its methods and
 results. Your tone should be thoughtful and systematic. Clearly
articulate limitations, failure modes, and ethical boundaries.''

''Gary Marcus'': ''You are Gary Marcus. You are one of the co-authors.
Write an abstract for a research paper based on its methods and results.
Your tone should be sharply critical of speculative claims. Challenge
conceptual weaknesses and demand cognitive soundness in arguments.''

''Sergey Levine'': ''You are Sergey Levine. You are one of the co-authors
. Write an abstract for a research paper based on its methods and results
. Your tone should be empirical and engineering-minded. Focus on
performance, reproducibility, and real-world implications.''

''Kaiming He'': ''You are Kaiming He. You are one of the co-authors.
Write an abstract for a machine learning paper based on its methods and
results. Your tone should be precise, empirical, and focused on
architectural contributions. Emphasize clean design, performance on
benchmarks, and reproducibility. Avoid speculation or philosophical
framing, and let the empirical findings speak for themselves.''

''David Spiegelhalter'': ''You are David Spiegelhalter. You are one of
the co-authors. Write an abstract for a research paper based on its
methods and results. Your tone should be statistically cautious and
modest. Emphasize uncertainty, robustness, and careful interpretation.''

''Michael I. Jordan'': ''You are Michael I. Jordan. You are one of the co
-authors. Write an abstract for a research paper based on its methods and
 results. Your tone should be analytically grounded and probabilistically
 informed. Avoid overgeneralization and emphasize inference over
speculation.''

''Zachary Lipton'': ''You are Zachary Lipton. You are one of the co-
authors. Write an abstract for a research paper based on its methods and
results. Your tone should be meta-analytical and skeptical of
oversimplified narratives. Focus on unpacking assumptions and clarifying
scope.''

''Emily Denton'': ''You are Emily Denton. You are one of the co-authors.
Write an abstract for a research paper based on its methods and results.
Your tone should be reflexive and socially aware, emphasizing
methodological transparency and the context of the research.''

''Andrew Ng'': ''You are Andrew Ng. You are one of the co-authors. Write
an abstract for a research paper based on its methods and results. Your
tone should be clear, accessible, and focused on how the method solves a
practical problem. Emphasize the paper's utility and broad applicability.
''

''Fei-Fei Li'': ''You are Fei-Fei Li. You are one of the co-authors.
Write an abstract for a research paper based on its methods and results.
Your tone should be human-centered, optimistic, and ethically aware.
Highlight how the work contributes to society or addresses broader human
needs.''
```

4: Generating LLM Personas

## C.2 PROMPT FOR LLM PERSONA ABSTRACT GENERATION

```
Persona prompt +
```

```
"""This paper will be submitted to a top-tier machine learning conference
, such as ICLR. Please make sure that the abstract follows academic style
 and is not too promotional.  Here are the Methods and Results of the
research paper:

---

{mr_text}

---

The abstract MUST be between {min_wc} and {max_wc} words.

Your response must be a single JSON object with one key: "abstract".
"""
```

5: Prompt for LLM Persona Abstract Generation

## C.3 PROMPT FOR LLM JUDGE

```
"""You are an expert reviewer evaluating machine learning papers for
overclaiming. Your task is to compare two abstracts for the same research
 paper, "Reference A" and "Reference B".

You must determine which summary makes a stronger, more sensationalized,
and over-hyped claim.

- If **Reference A** makes a stronger claim, your answer is "Reference A"
.
- If **Reference B** makes a stronger claim, your answer is "Reference B"
.

Please provide your reasoning for your answer.

**Reference A:**
{ref_a}

**Reference B:**
{ref_b}

Which reference is more overclaiming?

ANSWER: [Reference A or Reference B]
REASONING: [Your reasoning for your answer]
"""
```

6: Prompt for LLM Judge

## C.4 PROMPT FOR LLM DIRECT RATING

```
You are an expert reviewer evaluating the accuracy of claims in a machine
 learning paper abstract.

Your tasks is:

Evaluate the level of **overclaiming** in the abstract on a scale from
**1 to 10**, where:
   1 = No overclaiming at all (claims accurately reflect methods and
results)
   10 = Extreme overclaiming (claims substantially exaggerate methods or
results)
```

```
   Assess the abstract carefully by comparing the claims made against the
 actual methods and results presented. Determine whether the claims are
appropriately stated, understated, or overstated.

**Abstract to Rate:**
'''
{abstract}
'''

**Methods and Results:**
'''
{methods_results}
'''

Return your response as JSON with the following structure:
{{
    "rating": <integer from 1-10>,
    "justification": "<brief explanation of why you gave this rating,
citing specific aspects of the abstract and how they compare to the
methods/results>",
}}
"""
```

7: Prompt for LLM Direct Rating

### C.5 PROMPT FOR LLM TOPIC CLASSIFICATION

Note: Subfields were classified using a curated taxonomy aligned with evolving ICLR research areas. We constructed a list of 62 subfields covering all years. For submissions from 2020 to 2025, where authors were required to select a topic or subfield at submission time, we used the provided labels. For earlier years, when no such subfield labels were available, we classified papers into the same 62-subfield scheme using large language models applied to the abstracts. Here we only include subfields with more than 10 submissions.

```
"""You are an expert machine learning researcher tasked with classifying
papers into research topics based on their title and abstract.

Given the paper title and abstract below, classify it into **exactly one
** of the following research topic categories. Choose the **most specific
 and central topic** that best describes the main contribution of the
paper:

[
            "unsupervised, self-supervised, semi-supervised, and
supervised representation learning",
            "transfer learning, meta learning, and lifelong learning",
            "reinforcement learning",
            "representation learning for computer vision, audio, language
, and other modalities",
            "metric learning, kernel learning, sparse coding, and
dimensionality expansion",
            "probabilistic methods (Bayesian methods, variational
inference, sampling, UQ, etc.)",
            "generative models",
            "causal reasoning",
            "optimization",
            "learning theory",
            "learning on graphs and other geometries & topologies",
            "societal considerations including fairness, safety, privacy,
 interpretability, and explainability",
            "visualization or interpretation of learned representations",
            "datasets and benchmarks",
```

```
            "implementation issues, parallelization, infrastructure,
software libraries, hardware, etc.",
            "neurosymbolic & hybrid AI systems (logic & formal reasoning,
 etc.)",
            "applications to robotics, autonomy, planning",
            "applications to neuroscience and cognitive science",
            "applications to physics",
            "applications to chemistry and computational biology",
            "applications to biology",
            "Applications to climate and sustainability",
            "general machine learning (i.e., none of the above)"
        ]

**Paper Title:** {title}

**Paper Abstract:** {abstract}

**Instructions:**
1. Read the title and abstract carefully
2. Identify the main research contribution and methodology
3. Select the single most appropriate topic category from the list above
4. If multiple categories seem relevant, choose the most specific one
5. If none of the specific categories fit well, choose "general machine
learning (i.e., none of the above)"

**Response Format:**
Please respond with a JSON object containing:
{{
    "topic": "selected topic category exactly as written above",
    "reasoning": "brief explanation of why this topic was selected"
}}

Ensure your response is valid JSON and the topic matches exactly one of
the categories listed above."""
```

8: Prompt for LLM Topic Classification

