# OpenReview forum: "Counterfactual LLM-based Framework for Measuring Rhetorical Style"
_ICLR.cc/2026/Conference — ICLR 2026 Poster_

### Official Review · Reviewer_sF21 · 2025-10-28

**Soundness:** 2
**Presentation:** 2
**Contribution:** 2
**Rating:** 2
**Confidence:** 4

**Summary:**

This work presents a counterfactual, LLM-based framework for measuring rhetorical style independently of content. It uses a calibrated panel of LLM personas to generate counterfactual texts, aggregates pairwise comparisons with a Bradley–Terry model, and infers rhetorical strength for new writings. Validated against human judgments, the method is robust to persona choice and produces fine-grained scores. Applied to ICLR submissions, it shows that rhetorical style predicts citations and media attention, revealing a notable increase in rhetorical strength since 2023.

**Strengths:**

1. This work introduces a counterfactual, LLM-based framework to measure rhetorical style independently of content.

2. It develops a calibrated panel of LLM personas to generate counterfactual writings, aggregates pairwise comparisons via a Bradley–Terry model, and provides a method to infer rhetorical strength for new texts.

3. The approach is validated against human annotations, is robust to persona choice, and produces fine-grained scores. Applied to a large dataset of ICLR submissions, the framework reveals that rhetorical style predicts citations and media attention, and that rhetorical strength has increased notably since 2023.

**Weaknesses:**

1. It remains unclear whether the Bradley-Terry score is sufficiently representative or effective for mimicking rhetorical style, especially given its low correlation with reviewer scores. Further analysis is needed (see my questions below).

2. The experimental section feels somewhat weak. There is no clear evidence of the reliability of the Bradley-Terry score, nor a detailed discussion of how it might generalize to broader research contexts.

3. The method depends on LLM-generated counterfactual abstracts and a calibrated panel of LLM judges. This setup introduces potential biases stemming from the choice of LLM personas and their training data, which may influence the assessment of rhetorical strength.

4. It is unclear whether the observed trends generalize beyond ICLR submissions, as all findings are drawn from a single conference domain.

**Questions:**

1. The selection of personas will largely affect the model performance. Therefore, it's important to elaborate how these personas are selected? Whether the LLMs strictly follow the personas.

2. According to Line 308, if rhetorical style were measured across full papers rather than just abstracts, would the correlation with peer-review scores increase, and could it then meaningfully predict reviewer evaluations?

3. If abstracts (Y) are used as a proxy for rhetorical style while the full paper content (X) represents the substantive content, could the limited scope of abstracts lead to an incomplete or biased estimation of Z? In other words, does using only abstracts risk conflating substantive content with rhetorical framing, since abstracts may omit key details present in X?

4. In Line 101, the authors compare the setup with several methods such as GAN, RLHF, DPO. It remains unclear to me how the setup is connected to this method. It would be helpful if the authors could elaborate more on it.

---

> ### Author Response · Authors · 2025-11-20
> **Response to Reviewer sF21**
>
> We sincerely thank the reviewer for engaging with our paper and for outlining both strengths and concerns. We appreciate these thoughtful comments and offer clarifications below. We would be glad to address any further questions the reviewer might have.
>
> ## Weakness 1:
> > 1. It remains unclear whether the Bradley-Terry score is sufficiently representative or effective for mimicking rhetorical style, especially given its low correlation with reviewer scores. Further analysis is needed (see my questions below).
>
> We appreciate this concern and agree that demonstrating reliability is essential. We would like to highlight the following points:
>
> (1) **The Bradley–Terry rhetorical score is highly predictive of real outcomes**
>
> Our analysis shows that rhetorical style strongly predicts downstream attention:
>
> - Citations: +24 citations per unit rhetorical increase (p < 0.001)
> - Media attention: significant, robust effect sizes
>
> These results persist even after controlling for review scores.
> This is direct evidence that the score captures a meaningful, behaviorally consequential dimension of scientific writing.
>
> (2) **The null correlation with review scores is expected and theoretically sensible**
>
> Reviewers evaluate the full paper rather than the abstract alone. In contrast, the broader scientific community and the media often engage with a paper primarily through its abstract, which helps explain why rhetorical style is strongly predictive of citations and media attention but not review scores.
>
> Additionally, rhetorical style is not part of the ICLR review criteria (which emphasize novelty, technical rigor, and empirical soundness). Therefore, the absence of a strong relationship with review scores aligns with the expectation that reviewers prioritize substantive contributions over rhetorical framing. The null result should not be interpreted as a weakness of the method; rather, it confirms that abstract-level rhetorical differences do not influence how reviewers evaluate papers' scientific contribution.
>
> ## Weakness 2:
> > 2. The experimental section feels somewhat weak. There is no clear evidence of the reliability of the Bradley-Terry score, nor a detailed discussion of how it might generalize to broader research contexts.
>
> In the **General Response**, we have added substantial new analyses cross-validating our rhetorical scores against the *LLM detection framework of Liang et al. (2024)*.
> 1. Validation of the Baseline: We first confirmed the Liang et al. estimator is accurate for our domain: it estimates essentially zero LLM usage ($\alpha=0.5\%$) for pre-LLM abstracts (2017–2023) and high usage ($\alpha=57.3\%$) for our known LLM-generated personas.
> 2. Strong Correlation with LLM Usage: In the 2024–2025 cohort, we observe a strong correlation ($r=0.904$) between our rhetorical scores and independent estimates of LLM usage. Papers in the highest rhetorical decile show 2.3$\times$ higher estimated LLM usage than those in the lowest decile (20.9% vs 9.0%).
> 3. Ruling out Confounders: We investigated whether high rhetorical scores simply "trick" the estimator (false positives).
>     - Control 1 (Human Text): For 2017–2023 papers (human-written), high rhetorical scores do not trigger higher LLM detection estimates; usage remains $<1\%$ across all rhetorical groups.
>     - Control 2 (LLM Text): Within our generated personas, higher rhetorical scores do not lead to higher detection rates ($r=0.04$).
>
> Conclusion: These controls demonstrate that our score is not measuring a feature that merely confuses detectors, but rather one that LLMs naturally generate (and that authors rely on LLMs to generate) when producing rhetorically elevated text.

---

> > ### Author Response · Authors · 2025-11-20
> > **Response to Reviewer sF21 - weakness 3 & 4**
> >
> > ## Weakness 3:
> > > 3. The method depends on LLM-generated counterfactual abstracts and a calibrated panel of LLM judges. This setup introduces potential biases stemming from the choice of LLM personas and their training data, which may influence the assessment of rhetorical strength.
> >
> > **Comment**: We appreciate the reviewer’s concern about potential bias arising from (i) LLM-generated personas and (ii) LLM judges. We address both components below.
> >
> > (1) **Bias in LLM judges is mitigated by pairwise calibration and human validation.**
> >
> > Our rhetorical measurement uses pairwise comparisons, not scalar ratings. Because the Bradley–Terry model relies on relative preferences, it is substantially insulated from global leniency/strictness or other systematic biases in the judge. What matters is only which abstract is judged to have stronger claims in each pair, not absolute score levels.
> >
> > This is precisely why Bradley–Terry models are widely used with noisy or imperfect annotators: consistent pairwise judgments yield a stable latent ranking even under individual bias.
> >
> > Importantly, we validated the judge directly with humans. Human annotators show 88.4% agreement with the LLM judge, and the resulting persona scores correlate strongly with human-derived scores (Spearman correlation coefficient = 0.92). This demonstrates that the judge’s preferences align closely with human rhetorical perception.
> >
> > (2) **Bias in persona selection is mitigated by broad coverage and subset stability.**
> >
> > The key requirement for our measurement framework is that the persona panel collectively spans a wide and meaningful range of rhetorical styles. The specific choice of any single persona is not important; what matters is that the set as a whole produces texts that differ substantially in rhetorical intensity. The Bradley–Terry calibration then positions real abstracts relative to this spectrum.
> >
> > While personas are hand-designed, they intentionally span a wide rhetorical spectrum. Their win rates against real abstracts vary from 13.5% to 94%, indicating substantial stylistic diversity rather than collapse onto an LLM-specific mode.
> >
> > We further conducted a subset-stability test: splitting the 30 personas into two disjoint groups of 15 and recalibrating the system independently. The resulting BT scores were highly consistent (Spearman ρ = 0.89), showing the measurement is not sensitive to which specific personas are choosed.
> >
> > (3) We acknowledge that persona construction is not yet systematic, and we plan to address this in follow-up work.
> >
> > While our current approach produces a stable rhetorical scale (as demonstrated by robustness-to-subset tests with mean Spearman 0.89), we agree that persona construction can be improved. A more principled approach, such as learning personas via methods like representation-space interpolation or chain-based sampling along structured rhetorical dimensions, would yield a more systematic and reproducible rhetorical spectrum. We view this as an important direction for future research and plan to explore these methods in follow-up work.
> >
> > Overall, the evidence indicates that neither the LLM judge nor the persona panel introduces systematic distortions affecting measurement reliability.
> >
> > ## Weakness 4:
> > > 4. It is unclear whether the observed trends generalize beyond ICLR submissions, as all findings are drawn from a single conference domain.
> >
> > **Comment:** We appreciate the reviewer’s question about generalizability. While our empirical analysis focuses on ICLR, we emphasize that **this choice reflects data availability rather than a limitation of the framework itself**.
> >
> > (1) the measurement framework is **domain-agnostic**. It operates purely on:
> > 1. extracted substantive content,
> > 2. persona-based counterfactual generations, and
> > 3. pairwise rhetorical comparisons aggregated via the Bradley–Terry model.
> >
> > None of these components rely on conventions unique to ICLR.
> >
> > (2) ICLR is a practical and well-justified testbed:
> >
> >  - It provides a large, multi-year corpus with consistent formatting.
> >  - **It includes both accepted and rejected submissions, without selection bias.**
> >  - **It offers publicly available peer-review ratings**, which give us a domain-internal control for substantive quality, and allow us to validate whether our rhetorical measurement is disentangled from content.
> >
> > (3) We expect the qualitative trends (e.g., associations between rhetorical style and downstream attention; the post-2023 increase linked to LLM-assisted writing) to extend to other ML conferences such as NeurIPS and ICML, which share similar writing norms, audiences, and review structures.
> >
> >
> > (4) Our focus on ICLR is intentional: the field is experiencing rapid growth and heightened attention, making it an important setting in which to understand how rhetorical presentation interacts with downstream attention. Broader application is certainly possible, though outside the scope of this current empirical analysis.

---

> > > ### Author Response · Authors · 2025-11-20
> > > **Response to Reviewer sF21 - question 1 & 2**
> > >
> > > ## Question 1:
> > > > 1. The selection of personas will largely affect the model performance. Therefore, it's important to elaborate how these personas are selected? Whether the LLMs strictly follow the personas.
> > >
> > > We appreciate the reviewer’s question about persona selection and adherence. We offer both a conceptual justification and empirical evidence.
> > >
> > > (1) How personas were selected (design rationale).
> > >
> > > Our personas were deliberately constructed to span a wide rhetorical spectrum, from highly cautious to highly visionary framings. In practice, we generated an initial set of candidate personas and retained those whose win-rates against real abstracts covered the full range of rhetorical strength (from ~13% to ~94%). This design ensures broad and approximately uniform coverage across rhetorical intensities.
> > >
> > > Note that the key requirement for our measurement framework is that the persona panel collectively spans a wide and meaningful range of rhetorical styles. The specific choice of any single persona is not important; what matters is that the set as a whole produces texts that differ substantially in rhetorical intensity. The Bradley–Terry calibration then positions real abstracts relative to this spectrum.
> > >
> > > (2) Whether the LLMs follow the personas.
> > >
> > > Empirically, we observe that the LLM follows these personas consistently and in predictable ways:
> > >  - The Bradley–Terry scores are highly stable and interpretable.
> > >  - In human validation (Section 4.2), human annotators agree with the LLM-judge’s pairwise comparisons 88.4% of the time, and persona scores correlate strongly with human-derived scores (Spearman ρ = 0.92).
> > >
> > > Together, these results demonstrate that the LLM follows persona instructions in a way that yields human-recognizable and stable rhetorical differences.
> > >
> > > (3) Model performance is not sensitive to the particular choice of personas.
> > >
> > > We explicitly tested this by splitting the 30 personas into two disjoint groups of 15 and recalibrating the full system separately. The resulting Bradley–Terry scales remained highly correlated (Spearman ρ = 0.89), showing that the estimation is robust to which personas are used.
> > >
> > >
> > >
> > > ## Question 2:
> > > > According to Line 308, if rhetorical style were measured across full papers rather than just abstracts, would the correlation with peer-review scores increase, and could it then meaningfully predict reviewer evaluations?
> > >
> > > **Comment:** We thank the reviewer for this thoughtful question. We agree that full-paper rhetorical style could, in principle, behave differently from abstract-level rhetoric. However, based on both theoretical considerations and our empirical findings, we do not expect full-paper measurement to meaningfully increase the correlation with reviewer scores.
> > >
> > > (1) Would the correlation likely increase if we measured full-paper rhetorical style?
> > >
> > > We believe it is not very likely. Reviewer evaluations at ICLR focus on novelty, technical rigor, empirical soundness, and clarity of exposition. These criteria pertain almost entirely to the substantive portions of the paper (method, theory, experiments). Rhetorical framing, whether in the abstract or the introduction, is not part of the review criteria and is not expected to affect reviewer scoring.
> > >
> > > Our empirical findings reinforce this interpretation: rhetorical strength predicts downstream attention (citations, media attention) but shows no effect on peer-review scores. This aligns with the expectation that reviewers prioritize content over rhetorical presentation.
> > >
> > > For this reason, even if we were to compute section-level rhetorical scores across the entire paper, we would not expect a substantial increase in correlation with reviewer ratings.
> > >
> > > (2) Why we focus on abstracts rather than full papers.
> > > Abstracts offer a standardized and comparable rhetorical unit across submissions. Full papers vary widely in structure (e.g., math-heavy vs. empirical papers), length, and narrative flow. Measuring rhetoric across full papers would introduce structural confounds and make it harder—not easier—to separate rhetorical style (Z) from content (X).
> > >
> > > (3) Abstract-level rhetoric is precisely where rhetorical effects matter most.
> > >
> > > The abstract is the only section every reviewer reads and the primary component engaged by the broader scientific audience and the media. This explains why rhetorical style is strongly predictive of citations and attention but not reviewer scores.
> > >
> > > While full-paper analysis is conceptually possible, it is not necessary for interpreting the present findings, nor do we expect it to change the core result: rhetorical framing influences community attention, not necessarily peer review evaluations.

---

> > > > ### Author Response · Authors · 2025-11-20
> > > > **Response to Reviewer sF21 - question 3**
> > > >
> > > > ## Question 3:
> > > > > 3. If abstracts (Y) are used as a proxy for rhetorical style while the full paper content (X) represents the substantive content, could the limited scope of abstracts lead to an incomplete or biased estimation of Z? In other words, does using only abstracts risk conflating substantive content with rhetorical framing, since abstracts may omit key details present in X?
> > > >
> > > >
> > > > **Comment:** We thank the reviewer for raising this important conceptual question. We agree that, in principle, rhetorical markers can appear throughout a full paper. However, using abstracts as the unit of analysis does not introduce bias into the estimation of rhetorical style Z. In fact, abstracts are the safest and least confounded location for estimating Z. We clarify below.
> > > >
> > > > (1) Using abstracts reduces, not increases, the risk of conflating content (X) with rhetorical style (Z).
> > > >
> > > > Full papers vary substantially in structure and length across subfields, and contain heterogeneous content (methods, proofs, experiments) that naturally differs in density and exposition. These differences introduce strong confounds: some areas require heavy math exposition, others require long empirical descriptions, etc. Measuring rhetorical style at the full-paper level risks capturing structural and content-driven differences rather than rhetorical framing.
> > > >
> > > > In contrast, abstracts are a highly standardized and comparable rhetorical unit:
> > > > - similar length,
> > > > - similar communicative purpose,
> > > > - similar conventions across subfields.
> > > >
> > > > This standardization helps isolate rhetorical framing from substantive content, precisely the goal of our counterfactual design.
> > > >
> > > > (2) Abstracts encode “surface-level framing,” which is where rhetorical variability is most concentrated.
> > > >
> > > > Rhetorical choices, including assertiveness, visionary framing, claims about significance, future impact, are primarily expressed in the abstract and introduction. The methodology or experiment sections of the full paper are far more constrained and contain less rhetorical flexibility. Thus, the abstract is the section where Z is most visible and most comparable across papers.
> > > >
> > > >
> > > > (3) Using abstracts does not bias the estimation of rhetorical style Z.
> > > >
> > > > Rhetorical style Z concerns how authors frame and present the contributions, not the technical depth or completeness of the paper. **The abstract already contains the rhetorical choices that matter most**: how strongly contributions are emphasized, how novel or impactful results are described, and how assertively claims are made.
> > > >
> > > > Because stylistic decisions Z are independent of the deep technical content, omitting those details does not distort the measurement of rhetorical style. In fact, restricting analysis to abstracts reduces confounding: abstracts are structurally comparable across papers, whereas full papers vary widely in organization, length, mathematical density, and subfield conventions.
> > > >
> > > > Finally, our counterfactual generation process holds the substantive content constant within the abstract domain, ensuring that variation in the generated texts reflects only rhetorical framing Z. This makes the abstract the appropriate unit for isolating and comparing rhetorical style.
> > > >
> > > > (4) Empirically, our estimator behaves as expected if Z is cleanly separated from X.
> > > >
> > > > Two key results support that we are not conflating X and Z:
> > > > - Z predicts citations and media attention, outcomes strongly associated with rhetorical framing.
> > > > - Z does not predict reviewer scores, which depend on the technical content X.
> > > >
> > > > This pattern is exactly what the theory predicts if Z is measured cleanly and independently of X.
> > > >
> > > > For these reasons, we believe abstract-level analysis provides the most principled and least confounded estimate of rhetorical style, and focusing on full papers would introduce more structural noise rather than reduce bias.

---

> > > > > ### Author Response · Authors · 2025-11-20
> > > > > **Response to Reviewer sF21 - question 4**
> > > > >
> > > > > ## Question 4:
> > > > > > 4. In Line 101, the authors compare the setup with several methods such as GAN, RLHF, DPO. It remains unclear to me how the setup is connected to this method. It would be helpful if the authors could elaborate more on it.
> > > > >
> > > > > **Comment:** We appreciate the reviewer’s question. The paragraph draws a high-level conceptual analogy across two relate, but importantly different, families of methods.
> > > > >
> > > > > (1) Our framework resembles GANs conceptually
> > > > >
> > > > > GANs involve:
> > > > > - a generator that produces outputs, and
> > > > > - a discriminator that evaluates those outputs.
> > > > >
> > > > > Our framework resembles this in high-level structure:
> > > > > - the personas act like multiple “generators,” producing alternative texts from the same content;
> > > > > - the LLM judge plays the role of a “discriminator,” comparing rhetorical strength.
> > > > >
> > > > > The key diffierence is that GANs involve adversarial training, whereas our method focuses on measuring rhetorical differences.
> > > > >
> > > > > (2) Our Bradley–Terry preference model is similar to RLHF/DPO
> > > > >
> > > > > RLHF and DPO:
> > > > > - take multiple outputs from the same input,
> > > > > - obtain preference comparisons, and
> > > > > - infer a latent preference model.
> > > > >
> > > > > This is much closer to what we do:
> > > > > - we generate multiple texts from the same content (via personas),
> > > > > - obtain pairwise preference judgments from an LLM judge,
> > > > > - fit a Bradley–Terry model to infer a latent rhetorical dimension.
> > > > >
> > > > > (3) Why both families were mentioned
> > > > >
> > > > > We mention both to help readers situate our method within a broader and well-established paradigm of using pairwise preferences to infer latent attributes. Although our pipeline differs fundamentally in purpose and does not involve adversarial training or policy optimization, the high-level structure is similar in that multiple outputs are generated from the same content and evaluated through comparative judgments. The analogy is intended to orient readers, not to imply methodological equivalence.
> > > > >
> > > > >
> > > > > We hope our response and new experiments answer your questions. We would appreciate it if you could re-evaluate the score for our paper. Your constructive feedback has made our paper much more rigorous.

---

> ### Comment · Reviewer_sF21 · 2025-11-26
> **Rebuttal Response**
>
> Thank you for your response. Most of my concerns are resolved. I've increased the score.

---

> > ### Author Response · Authors · 2025-11-27
> > **New Robustness Analysis addressing your concern on Reliability**
> >
> > Dear Reviewer sF21,
> >
> > We sincerely thank you for engaging with our rebuttal and for raising your score. We really appreciate your reassessment.
> >
> > We understand from your previous comments that you may still have reservations regarding the **reliability** of the rhetorical scoring. To address this directly, we have just posted a **General Response** detailing a new **scaling analysis**.
> >
> > We systematically **varied the persona set size from 1 to 30**. The results show that our scores are highly robust ($\rho > 0.8$) even when **using only 5 randomly selected personas**, and that the variance across random trials is negligible. We have included these details in the updated paper **marked in Blue**.
> >
> > **Since this analysis provides the quantitative evidence of reliability you requested, we respectfully ask if you would consider whether the paper now meets the bar for acceptance. Thank you so much!**

---

### Official Review · Reviewer_Kcz4 · 2025-10-28

**Soundness:** 3
**Presentation:** 3
**Contribution:** 2
**Rating:** 6
**Confidence:** 3

**Summary:**

This paper assumes that each paper abstract has a particular style of presenting the problem, methods and results, which they call the “rhetorical style” of the paper. The authors build a system that can predict the rhetorical score for a paper given its abstract. The method involves 30 different rewrite prompts that use LLMs to rewrite an abstract in different styles. These styles are assumed to be “counterfactuals”. The method is used for a series of ICLR paper submissions and the results show that, once controlling for review scores, the papers with a higher rhetorical score tend to be cited and discussed more.

**Strengths:**

I found the correlation between the rhetorical score and the “popularity” of a paper an interesting result, which may show how humans are biased by the presentation style.

**Weaknesses:**

I don’t see any serious weaknesses. Wondering what the practical application and implications are. Shall we all adopt a writing style that results in a high rhetorical score? :) Also not clear at all why the method is called “counterfactual”.

**Questions:**

It was not clear to me why the method is called “counterfactuals”. Not clear why the rewriting prompts are assumed to produce counterfactual abstracts. Could you please clarify this term choice? It seems to be inappropriate for this context.

What are the practical implications and/or applications of this technique?

---

> ### Author Response · Authors · 2025-11-20
> **Response to Reviewer Kcz4: Weakness 1**
>
> We sincerely thank the reviewer for their thoughtful assessment of our work. We appreciate the recognition of the empirical findings. We are grateful for the reviewer’s note that they “don’t see any serious weaknesses,” and we also appreciate the questions raised about our use of the term “counterfactual” and about the practical implications of the technique. These comments were very helpful, and we address each point in detail below.
>
> ## Weakness 1:
> >1. I don’t see any serious weaknesses. Wondering what the practical application and implications are. Shall we all adopt a writing style that results in a high rhetorical score? :) Also not clear at all why the method is called “counterfactual”.
>
> **Comment:** We appreciate the reviewer’s positive assessment and the encouraging note that no serious weaknesses were identified. We also thank the reviewer for raising two important clarification questions regarding (1) the meaning of “counterfactual” in our method, and (2) the practical implications of the technique.
>
> (1) “Counterfactual” means "what-if"
>
> **Definition:** adapted from causal inference, a counterfactual refers to an unobserved scenario: it is the outcome that would have happened if a specific event (or "treatment") had been different from what factually occurred.
>
> In scientific writing, we never observe how an author would have written the exact same technical content had they chosen a different rhetorical stylem, e.g., more cautious, more assertive, or more visionary. **That unobserved alternative is precisely the counterfactual.**
>
> Our framework constructs structured approximations of these unobserved alternatives using LLM personas. Each persona is given the same extracted substantive content of a paper and generates an abstract that reflects a distinct rhetorical style. In this design, the underlying technical content is held constant, while the rhetorical framing varies.
>
> Thus, **each persona-generated abstract represents a counterfactual answer to a “what-if” question**:
> - “What would this abstract look like if the author had written it in a more cautious style?”**
> - “What if the same content were framed more assertively, or more visionary?”**
>
> This mirrors the structure of a counterfactual in causal inference:
> - The unit is the content of the paper.
> - The treatment is the rhetorical style.
> - The counterfactual is the version of the abstract under a different rhetorical “treatment,” with the underlying scientific contribution held constant.
>
> The term is not meant to evoke causal identification in the strict econometric sense, but rather **the conceptual idea of comparing what actually was written to what could have been written, had a different rhetorical choice been made.**
>
> (2) Practical implications
>
> First, while our results show that rhetorical style is associated with greater attention, we do not advocate for hype or exaggerated framing. **Our goal is descriptive, not normative: to measure how rhetorical choices shape how scientific work is perceived.** Writing clearly and compellingly is important, but exaggeration or distortion is not aligned with scientific communication norms, and our framework should not be interpreted as encouraging such behavior.
>
> Second, our current study examines short- and medium-term outcomes such as media attention and citations. We agree that these are only part of the picture. In future work, we plan to investigate longer-term consequences, including:
> - effects on researchers’ publication and career trajectories,
> - reputational consequences of consistently using highly rhetorical framing,
> - whether rhetorical strategies have diminishing returns or potential downsides over a career.
>
> This broader inquiry is necessary to understand whether rhetorical elevation reflects healthy communication practices or could incentivize unsustainable “hype cycles” in the field.
>
> Finally, regarding practical applications: our framework is designed to measure rhetorical style, not prescribe it. It can help conferences, journals, and authors understand how style and substance interact, but it is not intended as a recipe for authors to artificially inflate rhetoric, especially now that we provide a principled way to quantify and monitor it.

---

> > ### Comment · Reviewer_Kcz4 · 2025-11-21
> >
> > Thank you for your clarification. I think some of it should make it into the paper as well. I will maintain my scores.

---

> > > ### Author Response · Authors · 2025-11-21
> > > **Re: Replying to Response to Reviewer Kcz4**
> > >
> > > We thank the reviewer for the prompt response and the positive engagement.
> > >
> > > We are glad that our response clarified the terminology and implications. Per your suggestion, **we have incorporated these clarifications** into the paper. Specifically, we added the following definition to the Introduction section (p2):
> > > >Counterfactual design. Conceptually, our design parallels a counterfactual framework in causal inference, representing "what-if" alternative outcomes under different scenarios. We ask how the evaluation of a paper would change if the rhetorical style were altered while the underlying substantive content remained the same. By coupling controlled generation with pairwise comparisons, we provide a scalable and principled instrument for analyzing how ML research is evaluated.
> > >
> > > Additionally, to further address your question regarding practical implications, we have added substantial **new experiments showing a strong correlation between rhetorical scores and the estimated LLM usage**: In the 2024–2025 cohort, we observe a strong correlation ($r=0.904$) between our rhetorical scores and independent estimates of LLM usage. Papers in the highest rhetorical decile show 2.3$\times$ higher estimated LLM usage than those in the lowest decile (20.9% vs 9.0%).
> > >
> > > Since the reviewer previously noted there were "no serious weaknesses" and that the questions regarding terminology have now been resolved, we respectfully ask if the reviewer would consider raising the score. We believe a higher score would better reflect the reviewer's assessment of the paper’s soundness and the resolution of the clarification points. Thank you for helping us improve the manuscript. We hope our findings can be shared with the ICLR community.

---

> ### Author Response · Authors · 2025-11-20
> **Response to Reviewer Kcz4: Questions**
>
> ## Questions:
> > 1. It was not clear to me why the method is called “counterfactuals”. Not clear why the rewriting prompts are assumed to produce counterfactual abstracts. Could you please clarify this term choice? It seems to be inappropriate for this context.
> > 2. What are the practical implications and/or applications of this technique?
>
> We thank the reviewer for these thoughtful questions. Both points, the meaning of “counterfactual” in our framework and the practical implications of the technique, are addressed in detail in our response to the above section. In brief:
>
> - We use the term “counterfactual” because each persona abstract answers a “what if” question: what if the exact same substantive content were expressed in a different rhetorical style? This allows us to isolate style from content.
> - Regarding practical implications, our framework is intended as a measurement tool for understanding how rhetorical style interacts with scholarly communication, not as prescriptive guidance for authors to increase rhetorical intensity.

---

> ### Author Response · Authors · 2025-11-27
> **Updated paper with revisions and an additional scaling analysis**
>
> Dear Reviewer Kcz4,
>
> Thank you again for your positive assessment and for noting previously that you saw "no serious weaknesses."
>
> We wanted to let you know that, in addition to clarifying the "counterfactual" terminology as requested, we have uploaded a revised PDF containing two major additions that further strengthen the paper's empirical rigor:
> 1.  A **scaling analysis** showing our scores are highly stable even with small, random persona subsets.
> 2.  An **LLM-usage analysis** showing a strong link ($r=0.904$) between LLM adoption and rhetorical style.
>
> **Given that we have addressed your terminology questions and further bolstered the empirical results, we respectfully ask if you would consider raising your score to stronger support the acceptance of this work. Many thanks!**

---

### Official Review · Reviewer_3KCn · 2025-11-01

**Soundness:** 3
**Presentation:** 3
**Contribution:** 3
**Rating:** 8
**Confidence:** 3

**Summary:**

This paper introduces a counterfactual framework powered by LLM that automatically measures the rhetorical style of research papers independent of its substantive content. The framework models rhetorical style as a one-dimensional variable which is independent to the substantive content of the paper, which jointly derive the surface text form of the paper. To measure this rhetorical style variable, the paper proposes to 1) extract objective and descriptive content of the paper; 2) steer the LLM with a wide spectrum of persona to generate diverse abstracts with counterfactual writings; 3) use LLM-as-judge to provide Bradly-Terry based pair-wise comparison for estimating the rhetorical strength score for each persona to form a spectrum of rhetorical strength; 4) situate the actual (query) abstract in this spectrum to obtain the rhetorical strength estimation for the real abstract. With this framework, the authors estimates the rhetorical strength of paper abstracts of ICLR submissions from 2017 to 2025 and conduct statistical analysis of the correlation between the strength and review scores/downstream attention. The results showcases that 1) the rhetorical estimation by the proposed framework is more effective than baseline methods; 2) while the estimated rhetorical strength showcase minimal correlation with review scores, stronger rhetorical style does lead to larger downstream attention,

**Strengths:**

1. The proposed framework address the issue of entanglement of substantive content with rhetorical style, overcomes the challenges of biased measurement of rhetorical style in prior work.
2. With the multi-persona counterfactual generation & Bradley-Terry scoring approach, the proposed framework yield high quality and less biased estimation of rhetorical scores than prior approaches.
3. The analysis of the predictive power of rhetorical scores on peer-review scores and downstream impact/attentions provide insightful findings of how different rhetorical styles affect the recognition of the work by the community.

**Weaknesses:**

1. The single-dimension formulation of the rhetorical strength measurement might have made an over-simplified assumption. For instance, the strength of rhetorical style might be multi-faceted: a paper might argue significant generalizability of their contributions and simultaneously put less emphasis of the novelty/impact. I am thus concerned if the single-dimension rhetorical strength could capture such variability.

**Questions:**

Please see the weaknessess above

---

> ### Author Response · Authors · 2025-11-20
> **Response to Reviewer 3KCn**
>
> We sincerely thank the reviewer for the positive and thoughtful assessment of our paper. We appreciate the recognition of the contributions of our counterfactual framework, especially its ability to disentangle rhetorical style from substantive content and to improve measurement quality over prior approaches. We are grateful for the reviewer’s evaluation that the paper is sound, clearly presented, and a strong candidate for acceptance.
>
> ## Weakness 1:
> > 1. The single-dimension formulation of the rhetorical strength measurement might have made an over-simplified assumption. For instance, the strength of rhetorical style might be multi-faceted: a paper might argue significant generalizability of their contributions and simultaneously put less emphasis of the novelty/impact. I am thus concerned if the single-dimension rhetorical strength could capture such variability.
>
> **Comment:** We appreciate the reviewer’s insight that rhetorical style is, in reality, multidimensional. We fully agree. Scientific writing involves many distinct stylistic dimensions: claims about generalizability, emphasis on novelty, framing of impact, degrees of uncertainty, narrative tone, and more. Each of these could, in principle, be modeled as its own axis of rhetorical variation.
>
> In this work, however, we treat rhetorical strength as a single latent dimension for two principled reasons.
>
> (i) Our primary methodological question is whether rhetorical framing, in aggregate, can be cleanly disentangled from substantive content at all. Establishing this first-order identifiability is clearest in a one-dimensional setting.
>
> (ii) Many consequential downstream behaviors (e.g., citations, media attention, perceived “hype”) respond to the overall rhetorical intensity of the presentation, which is naturally captured by a unidimensional index. This composite “rhetorical strength” aligns with how readers often form holistic impressions when encountering an abstract.
>
> Importantly, our use of a single dimension is not a claim that rhetorical style is inherently one-dimensional, but rather a pragmatic and principled choice for establishing a baseline measurement. The framework itself is fully extensible: by modifying the judging criteria, one could derive multidimensional rhetorical scores, for example, capturing distinct dimensions such as novelty emphasis, generalizability claims, uncertainty hedging, and beyond.

---

> ### Author Response · Authors · 2025-11-27
> **Additional experiments and updated paper**
>
> Dear Reviewer 3KCn,
>
> Thank you again for your strong support of our work! We wanted to let you know that, in response to feedback from you and other reviewers, we have uploaded a revised PDF containing two major additions that further strengthen the paper:
>
> 1.  A **scaling analysis** showing our scores are highly stable even with small, random persona subsets.
> 2.  An **LLM-usage analysis** showing a strong link ($r=0.904$) between LLM adoption and rhetorical style.
>
> We believe these additions further solidify the validity of the framework and the paper's claims. Many thanks for your support again!

---

### Official Review · Reviewer_WLet · 2025-11-03

[review text omitted: it was posted to a different submission]

---

> ### Author Response · Authors · 2025-11-12
> **Referring to wrong paper?**
>
> Dear Reviewer, this review appears to refer to another paper (possibly BiasRetriever). Could you please verify if this was an error? We would greatly welcome your insights on our paper once the correct review is available.

---

> > ### Comment · Reviewer_WLet · 2025-11-12
> >
> > Apologies to the authors. Multiple reviews, copied the wrong materials to this one. I've gone ahead and updated the reviews and scores. Glad to discuss.

---

> ### Author Response · Authors · 2025-11-20
> **Response to Reviewer WLet: Weakness 1**
>
> We sincerely thank the reviewer for your thoughtful and generous evaluation of our work.
>
> We are grateful for the reviewer’s recognition of the paper’s strengths, including the novelty of the counterfactual LLM-based measurement framework. The reviewer’s framings of our contribution as “genuinely original” and “as far as one can tell, novel” are deeply appreciated.
>
> We also thank the reviewer for raising several sharp questions. These comments were invaluable and directly motivated a set of new  experiments and analyses that have substantially improved the clarity and robustness of the paper. Below, we respond to each point in turn.
>
> ## Weakness 1:
> >1. Confounding interpretation unclear: Post-2023 increase could reflect (a) authors choosing stronger rhetoric OR (b) LLMs nudging authors toward LLM-preferred styles. Framework cannot disentangle these mechanisms since LLMs themselves exhibit rhetorical preferences—this is the paper's most significant limitation
>
> **Comment:** We appreciate the reviewer’s concern about the potential confounding between (a) authors independently choosing stronger rhetoric and (b) LLMs nudging authors toward LLM-preferred rhetorical styles. Our additional analyses help clarify these mechanisms.
>
> First, our evidence directly supports mechanism *(b) LLMs nudging authors toward more rhetorically elevated styles*. During persona construction, the LLM systematically produced stronger rhetorical text than the original human-written abstracts: two-thirds of personas achieved higher rhetorical scores than their corresponding human abstracts, and constructing genuinely low-rhetoric personas was significantly more challenging. This pattern indicates that LLMs have an inherent upward pull on rhetorical intensity.
>
> Second, our longitudinal analysis provides evidence against mechanism *(a) authors independently escalating rhetoric prior to LLM adoption*. If authors were gradually choosing stronger rhetoric due to competitive dynamics or field-wide stylistic shifts, we would expect a steady upward trajectory over time. However, Figure 3 in the paper shows the opposite: from 2018 to 2023, rhetorical scores exhibit a consistent downward trend. This pattern is inconsistent with the hypothesis that authors were already transitioning toward stronger rhetorical writing style before LLM tools became widespread.
>
> Taken together, the evidence aligns more strongly with the interpretation that **the post-2023 increase in rhetorical intensity reflects the adoption of LLM writing tools, which themselves tend to generate more rhetorically elevated language**.
>
> Finally, we acknowledge that some authors may have wanted to write in a stronger rhetorical style earlier and only gained the ability to do so with LLM assistance. Our study does not attempt to infer authors’ intentions; it focuses on observable shifts in writing behavior. Nonetheless, the patterns we document are most consistent with LLM-driven stylistic change rather than independent human-driven escalation.

---

> ### Author Response · Authors · 2025-11-20
> **Response to Reviewer WLet: Weakness 2**
>
> ## Weakness 2:
> >2. Ad-hoc persona construction: 30 hand-crafted personas lack systematic derivation. No validation that personas span human rhetorical space rather than just LLM preference distributions. Could benefit from comparison to simpler counterfactual baselines or human-written counterfactuals
>
> **Comment:** We appreciate the reviewer’s insight. We fully acknowledge that our current persona construction is hand-crafted and not yet derived through a systematic procedure. Below we clarify what we have validated, what limitations remain, and how we plan to improve the persona-generation process in future work.
>
> (1) Coverage and span of personas.
>
> The key requirement for our measurement framework is that the persona panel collectively spans a wide and meaningful range of rhetorical styles. The specific choice of any single persona is not important; what matters is that the set as a whole produces texts that differ substantially in rhetorical intensity. The Bradley–Terry calibration then positions real abstracts relative to this spectrum.
>
> Although our personas are hand-designed, we constructed them with explicit attention to coverage. In particular, we targeted personas whose generated texts achieve approximately uniform win rates against the original abstracts, ensuring broad representation across the rhetorical distribution. As shown in Figure 2 of the main paper, persona win rates vary widely: from 94% (highly assertive, visionary styles) to 13.5% (highly cautious, hedged styles). This wide spread indicates that the panel indeed spans the empirical rhetorical space we observe in ICLR abstracts and provides a robust basis for calibrating the latent rhetorical scale.
>
> (2) Human validation indicates personas capture human rhetorical space.
>
> To assess whether persona differences reflect variation recognizable to human readers rather than merely LLM-specific artifacts, we conducted a human evaluation (Section 4.2). Humans exhibit 88.4% agreement with LLM-judge comparisons and the Bradley–Terry persona scores correlate strongly with human-derived scores (Spearman ρ = 0.92). This suggests that personas do indeed reflect rhetorical attributes perceptible to humans, not solely LLM idiosyncrasies.
>
> (3) We acknowledge that persona construction is not yet systematic, and we plan to address this in follow-up work.
>
> While our current approach produces a stable rhetorical scale (as demonstrated by robustness-to-subset tests with mean Spearman 0.89), we agree that persona construction can be improved. A more principled approach, such as learning personas via methods like representation-space interpolation or chain-based sampling along structured rhetorical dimensions, would yield a more systematic and reproducible rhetorical spectrum. We view this as an important direction for future research and plan to explore these methods in follow-up work.

---

> ### Author Response · Authors · 2025-11-20
> **Response to Reviewer WLet: Weaknesses 3 and 4**
>
> ## Weakness 3:
> >3. Abstract-only measurement scope: Pragmatically justified but limits claims about full-paper rhetoric and how reviewers actually evaluate submissions, potentially explaining the null correlation with review scores
>
> **Comment:**
>
> We agree that focusing on abstracts constrains the scope of our claims. Our framework is intentionally designed to measure rhetorical style in the portion of a paper that is most comparable across submissions and most consistently LLM-assisted. We clarify a few key points:
>
> (1) Abstracts are the section where rhetorical style is most standardized and most influential.
>
> The abstract is the one component that every reviewer reads, regardless of expertise level or review workload, and it serves as the entry point to forming expectations about a submission. Thus, even though we do not analyze full papers, abstract-level rhetoric remains highly meaningful for understanding scientific communication.
>
> (2) Abstract-level measurement avoids confounds present in full papers.
>
> Full-paper rhetoric is influenced by differing structures across subfields, heterogeneous section lengths, and varying mathematical density. Measuring rhetoric across full papers introduces substantial noise. By focusing on abstracts, we ensure comparable rhetorical units across papers and isolate rhetorical style from content structure.
>
> (3) **The null correlation with review scores is expected and theoretically sensible**.
>
> Reviewers evaluate the full paper rather than the abstract alone. In contrast, the broader scientific community and the media often engage with a paper primarily through its abstract, which helps explain why rhetorical style is strongly predictive of citations and media attention but not review scores.
>
> Additionally, rhetorical style is not part of the ICLR review criteria (which emphasize novelty, technical rigor, and empirical soundness). Therefore, the absence of a strong relationship with review scores aligns with the expectation that reviewers prioritize substantive contributions over rhetorical framing. The null result should not be interpreted as a weakness of the method; rather, it confirms that abstract-level rhetorical differences do not influence how reviewers evaluate papers' scientific contribution.
>
> (4) Extending the framework to full papers is possible but left for future work.
>
> We agree this is an important direction. Applying our approach to sections such as the introduction or conclusion could reveal additional patterns. This extension requires substantially more compute resources and hierarchical modeling of rhetorical structure, which we view as exciting future work beyond the scope of the current study.
>
> ## Weakness 4:
> >4. LLM judge validation incremental: 88.4% human agreement is strong but incremental given MT-Bench established GPT-4 achieves 80-85% alignment matching inter-human agreement (Zheng et al., 2023)
>
> **Comment:** We appreciate this observation. While MT-Bench demonstrates strong human–LLM alignment, our validation operates in a fundamentally different domain and serves a different purpose.
>
> (1) Our task focuses on scientific rhetorical style, not conversational quality.
>
> MT-Bench evaluates general dialogue, reasoning, and preference judgments, whereas our LLM judge performs fine-grained pairwise comparisons of scientific abstracts along a single latent rhetorical dimension (e.g., assertiveness, visionary framing, future orientation). These rhetorical distinctions are subtle, domain-specific, and structurally different from MT-Bench tasks. Thus, MT-Bench alignment cannot substitute for validation on our task.
>
> (2) Our human agreement is measured in the exact task we use in the pipeline.
>
> The 88.4% agreement reflects how consistently humans align with the LLM judge on scientific rhetorical comparisons, not on unrelated conversational benchmarks. This direct, task-matched validation confirms that the judge recognizes the same rhetorical variation as human readers.
>
> (3) **Our contribution is not the LLM judge itself, but how the judge integrates into a counterfactual framework.**
>
> Our use of the LLM judge is part of a broader methodological pipeline: counterfactual personas, pairwise comparisons, Bradley–Terry aggregation, and downstream analysis of rhetorical style. The validation shows the judge is sufficiently reliable to support this pipeline. Thus, the judge is not intended to improve upon prior LLM-evaluation work but to ensure that the entire framework is grounded in human-recognizable rhetorical differences. Our validation demonstrates exactly this.

---

> ### Author Response · Authors · 2025-11-20
> **Response to Reviewer WLet: Questions**
>
> ## Question 1:
> >1. LLM adoption vs. rhetorical intent. Can you directly compare your rhetorical scores against LLM-detection scores using existing frameworks (e.g., Liang et al., 2024)? Can you share the correlation and explain how your method captures the author's rhetorical choices beyond LLM usage patterns? This is essential to further establishing the framework's unique contribution.
>
> **Comment:** We appreciate the reviewer’s insightful question, which directly motivated several new experiments and analyses. In brief, we now (i) report the correlation between rhetorical scores and LLM-usage estimates, (ii) conduct new mechanism tests showing that strong rhetorical style alone does not cause the estimator to misclassify human-written text as LLM-generated, and (iii) provide evidence that the post-2023 rise in rhetorical style is more consistent with increased adoption of LLM writing tools than with independent shifts in author behavior. These additions substantially clarify the relationship between rhetorical style and LLM adoption and strengthen the contribution of our framework.
>
> For a complete discussion of these analyses, including the correlation results, pre-2024 baseline checks, persona-based tests, we refer the reviewer to our **General Response to All Reviewers**, where we present the full set of findings and details.
>
> ## Question 2:
> >2. Temporal mechanism disentanglement. The post-2023 increase could reflect competitive pressure, LLM tool adoption nudging style, or both. Can you differentiate these mechanisms? For example, do papers with high LLM-adoption signatures show different rhetorical patterns than low-adoption papers in 2024-2025? This confounding threatens the paper's primary temporal finding.
>
> **Comment:** We thank the reviewer for highlighting this important question. We provide evidence that the post-2023 rise in rhetorical style is more consistent with increased adoption of LLM writing tools than with independent shifts in author behavior.
>
> For a complete discussion of these analyses, including the correlation results, pre-2024 baseline checks, persona-based tests, we refer the reviewer to our **General Response to All Reviewers**, where we present the full set of findings and details.
>
> ## Question 3:
> >3. Reviewer correlation interpretation. The near-zero correlation with peer review deserves dedicated investigation. Have you examined: a) whether rhetorical style predicts review score variance across reviewers, b) full-paper rhetoric vs. abstract-only rhetoric, or c) non-linear relationships? Given your policy implications focus on peer review, this hypothesis finding should be explained.
>
> **Comment:** We thank the reviewer for raising this important question and for suggesting specific diagnostic checks. We conducted additional analyses targeting a) reviewer-level variation and c) potential non-linear relationships, and summarize the results below.
>
> a) Reviewer-level variance.
> For each paper, we computed the variance of review scores across reviewers and tested whether rhetorical strength predicts this variance. We found no meaningful relationship: rhetorical style does not explain differences in reviewer disagreement. This reinforces the conclusion that rhetorical framing in abstracts does not systematically influence reviewer evaluations.
>
> c) Non-linear effects.
> To examine potential non-linear relationships, we extended our baseline regression of reviewer score on rhetorical strength by adding a quadratic term (i.e., reviewer score = β₁·rhetorical + β₂·rhetorical²). The quadratic term was not significant. This indicates that rhetorical style does not exert a measurable influence on peer-review scores, either linearly or non-linearly.
>
> b) Full-paper rhetoric.
> We agree that analyzing full-paper rhetorical style is valuable, but it is beyond the scope of the present study. Measuring rhetoric across full papers introduces substantial heterogeneity in length, structure, and mathematical density, requiring a separate modeling pipeline. We now state explicitly that full-paper rhetorical analysis is an important direction for future work.
>
> The near-zero correlation with review scores is theoretically sensible. Reviewers evaluate the full submission, focusing on novelty, rigor, and empirical soundness—rather than rhetorical framing in the abstract. In contrast, abstract, level rhetoric strongly predicts downstream attention (citations, media), consistent with how broader scientific and public audiences consume research.

---

> ### Author Response · Authors · 2025-11-27
> **New Experiment disentangling LLM Usage from Rhetorical Style**
>
> Dear Reviewer WLet,
>
> We wanted to draw your attention to our **General Response**, where we detail a major new experiment we conducted in response to your question about the mechanism (LLM adoption vs. authorial intent).
>
> Using the Liang et al. (2024) detection framework, we found a strong correlation ($r=0.904$) between rhetorical score and estimated LLM usage. Crucially, we also confirmed that high-rhetoric *human-written* papers (2017-2023) are **not** falsely flagged by the detector. This provides direct evidence for the mechanism you hypothesized: LLM tools are indeed driving the rhetorical shift.
>
> We have updated the paper with these findings **marked in Blue**. Thank you for pushing us to investigate this; it has significantly strengthened the manuscript.
>
> **Since this new quantitative evidence directly resolves the potential confounding factor you highlighted, we respectfully ask if you would consider increasing your score to reflect the strengthened validity of the framework. Thank you so much!**

---

### Author Response · Authors · 2025-11-20
**General response**

# General response

We appreciate the reviewers' constructive feedback and insightful comments, which greatly help us improve our work. We are especially encouraged that the reviewers recognized the novelty and methodological strength of our approach, highlighting the “scale and rigor” of our counterfactual framework and its “robust validation.” We are also very greateful that the reviews commended our method for “overcoming the challenges of biased measurement” inherent in prior work, as well as our findings regarding the impact of rhetorical style on scientific popularity.

To directly address the question of whether our rhetorical scores simply reflect LLM usage patterns rather than capturing genuine author intent, we conducted additional experiments using the LLM detection framework of Liang et al. (2024). Our analyses reveal a strong correlation between LLM usage and rhetorical score.

## New Experiment: rhetorical score vs LLM usage

Liang et al. (2024)'s method analyzes statistical distributions of linguistic features across many documents to infer the proportion of LLM-generated text in a corpus (denoted as α). Their Maximum Likelihood Estimation (MLE) approach estimates LLM usage at the population level, not per-abstract. Therefore, to compare our per-abstract rhetorical scores against LLM usage estimates using Liang et al. (2024)'s method, we grouped abstracts by rhetorical score and estimated LLM usage for each group.

### Validation of Liang et al. (2024) estimator

We begin by validating the Liang et al. (2024) estimator on two dimensions. First, abstracts submitted in 2017-2023 ICLR cycle were drafted during a period when modern LLM-based writing tools were not yet widely adopted by researchers. As such, these abstracts serve as a natural ‘pre-LLM’ baseline, and their estimated LLM usage should be close to zero. Consistent with this expectation, we find that the estimated LLM usage during 2017-2023 is 0.5%, indeed very close to 0%.

Second, in our experiment all persona abstracts are known to be LLM-generated, so their estimated LLM usage should be substantially higher than zero. The average estimated α across persona groups is 57.3%. While this value is well below 100%, it is nevertheless far greater than zero, indicating that the estimator reliably distinguishes LLM-generated text from human-written text at the population level, even if it does not fully capture the true proportion in this synthetic setting.

### Findings: LLM Usage and rhetorical style are strongly correlated


We now examine the relationship between rhetorical scores and estimated LLM usage. Having validated that LLM usage is effectively zero in the 2017–2023 submissions, we restrict our analysis to the 2024–2025 cohort (n = 2,000). We divide these papers into 10 equally sized groups (each containing approximately 200 papers), based on their rhetorical scores, with each group representing 10% of the distribution. The correlation between group-level mean rhetorical score and group-level estimated LLM usage is strongly positive, with a Pearson correlation coefficient of 0.904.


Table 1. Estimated LLM Usage by Rhetorical Groups during 2024–2025
| **Group** | **Rhetoric Range**     | **Mean Rhetoric** | **Estimated LLM Usage (α)** |
|-----------|-------------------------|--------------------|------------------------------|
| 0 (lowest) | [-4.74, -2.95] | -3.534 | 9.0%  |
| 1          | [-2.95, -2.25] | -2.453 | 11.1% |
| 2          | [-2.25, -1.69] | -1.895 | 10.0% |
| 3          | [-1.69, -1.19] | -1.353 | 16.7% |
| 4          | [-1.19, -0.75] | -0.904 | 15.4% |
| 5          | [-0.75, -0.13] | -0.459 | 15.7% |
| 6          | [-0.13, 0.27]  | 0.019  | 18.5% |
| 7          | [0.27, 0.86]   | 0.518  | 17.9% |
| 8          | [0.86, 1.96]   | 1.323  | 17.6% |
| 9 (highest) | [1.96, 4.53]  | 3.138  | 20.9% |

Table 1 presents the estimated LLM usage in the 2024–2025 paper batch across the ten rhetorical groups. It reveals clear evidence of differential LLM adoption by rhetorical intensity. The pattern is strongly monotonic: **groups with higher mean rhetorical scores exhibit substantially higher estimated LLM usage.** For example, papers in the highest-rhetoric group (Group 9; mean score = 3.14) show an overall estimated LLM usage of α = 20.9%, which is approximately 2.3× higher than the lowest-rhetoric group (Group 0; mean score = –3.53; α = 9%). Intermediate groups follow the same upward gradient, with each increase in rhetorical intensity corresponding to progressively higher estimated LLM usage.

---

> ### Author Response · Authors · 2025-11-20
> **General response (2)**
>
> The observed correlation between rhetorical scores and estimated LLM usage could arise from two different mechanisms:
>  - Mechanism 1. LLMs may intrinsically produce more rhetorically elevated language.
>  - Mechanism 2. Texts with stronger rhetorical scores may be more likely detected as LLM-generated by the Liang et al. (2024) estimator.
>
> We investigate both possibilities.
>
> A rigorous assessment of whether LLMs intrinsically produce more rhetorically elevated language (Mechanism 1) would require access to the exact prompts and instructions human authors used when drafting their papers, but such information is typically not accessible. However, we can draw informative inferences from our own persona-generation process. When constructing personas, we aimed to span the empirical distribution of rhetorical styles observed in the original abstracts so that personas could serve as stylistic “anchors” across the rhetorical spectrum. In practice, **generating low-rhetoric personas proved more difficult than generating high-rhetoric ones.** Meanwhile, even when we explicitly attempted to balance personas by targeting uniform win rates against original abstracts, we observed systematic differences in the rhetorical intensity produced by different persona prompts. As shown in Figure 2 in the paper, **about 2/3 of the personas produce abstracts that outperform the original abstracts in rhetorical score**. This asymmetry suggests that, under comparable prompting conditions, **LLMs tend to generate more rhetorically elevated language than the human-written abstracts they are paired with**, providing indirect evidence consistent with Mechanism 1.
>
>
> To assess whether high rhetorical scores alone lead the Liang et al. (2024) estimator to classify text as LLM-generated (Mechanism 2), we conduct two complementary checks. First, we examine whether human-written texts with stronger rhetorical scores are more likely to be detected as LLM-generated. We apply the estimator to the 2017–2023 paper batch, which were written before the widespread availability of LLM writing tools, and divide these papers into 10 equal-sized groups based on rhetorical scores (each containing 10% of the papers).
>
> Table 2 summarizes the estimated LLM usage across these rhetorical groups. Across all groups, the estimated usage remains near zero. For example, even the highest-rhetoric group (Group 9) shows extremely low estimated LLM usage (0.5%). If rhetorical intensity mechanically inflated α, we would expect this group to exhibit much higher estimates. It does not. This pattern indicates that **strong rhetorical style alone does not cause the estimator to misclassify human-written text as LLM-generated**.
>
> Table 2. Estimated LLM Usage by Rhetorical Groups during 2017–2023
> | **Group** | **Rhetoric Range**     | **Mean Rhetoric** | **LLM Usage (α)** |
> |-----------|-------------------------|--------------------|--------------------|
> | 0 (lowest)         | [-4.74, -2.95]          | -3.725             | 0.3%               |
> | 1         | [-2.95, -2.25]          | -2.692             | 0.5%               |
> | 2         | [-2.25, -1.96]          | -2.113             | 0.4%               |
> | 3         | [-1.96, -1.43]          | -1.668             | 0.3%               |
> | 4         | [-1.43, -0.96]          | -1.272             | 0.7%               |
> | 5         | [-0.96, -0.72]          | -0.861             | 0.3%               |
> | 6         | [-0.72, -0.13]          | -0.400             | 0.8%               |
> | 7         | [-0.13, 0.46]           | 0.111              | 0.5%               |
> | 8         | [0.46, 1.27]            | 0.781              | 0.7%               |
> | 9 (highest)        | [1.27, 4.53]            | 2.164              | 0.5%               |

---

> > ### Author Response · Authors · 2025-11-20
> > **General response (3)**
> >
> > Second, we apply the Liang et al. (2024) estimator to our LLM-generated persona abstracts, grouped by persona. In this setting, all texts are known to be LLM-generated. Table 3 reports rhetorical scores and estimated LLM usage across personas. As shown, **personas with higher rhetorical scores do not consistently receive higher α estimates**. The correlation between persona-level rhetorical scores and estimated LLM usage is extremely weak (Pearson r = 0.04).
> >
> > Taken together, these findings indicate that **stronger rhetorical style does not inherently increase the likelihood that the Liang et al. estimator classifies text as LLM-generated**, effectively ruling out Mechanism 2. Thus, **the post-2023 rise in rhetorical style is more consistent with increased adoption of LLM writing tools than with independent shifts in author behavior**.
> >
> > Table 3. Estimated LLM Usage by Persona Groups
> > | **Persona** | **Rhetorical Score (Rank)** | **Estimated LLM Usage α (Rank)** | **Persona** | **Rhetorical Score (Rank)** | **Estimated LLM Usage α (Rank)** |
> > |-------------|------------------------------|--------------------------|-------------|------------------------------|---------------------------|
> > | "Fei-Fei" | 3.106 (1) | 0.727 (3) | "Emily" | 0.182 (16) | 0.636 (8) |
> > | "Pieter" | 2.687 (2) | 0.621 (10) | "Yejin" | 0.122 (17) | 0.702 (4) |
> > | "Ilya" | 2.374 (3) | 0.537 (21) | "Methods-Obsessed Researcher" | -0.029 (18) | 0.249 (30) |
> > | "Geoffrey" | 2.032 (4) | 0.641 (7) | "Michael" | -0.127 (19) | 0.551 (17) |
> > | "Yoshua" | 1.957 (5) | 0.642 (6) | "Teaching-Oriented Faculty" | -0.346 (20) | 0.457 (26) |
> > | "FAANG Researcher" | 1.531 (6) | 0.523 (22) | "Percy" | -0.717 (21) | 0.513 (23) |
> > | "Yann" | 1.280 (7) | 0.582 (15) | "Margaret" | -1.235 (22) | 0.610 (13) |
> > | "Sleep-Deprived PhD Student" | 0.976 (8) | 0.415 (28) | "Zitnick" | -1.422 (23) | 0.606 (14) |
> > | "Andrew" | 0.973 (9) | 0.618 (11) | "Zachary" | -1.674 (24) | 0.696 (5) |
> > | "Sergey" | 0.940 (10) | 0.540 (19) | "Overly Cautious Statistician" | -1.840 (25) | 0.572 (16) |
> > | "Mathematical Theorist" | 0.721 (11) | 0.613 (12) | "Gary" | -2.067 (26) | 0.779 (2) |
> > | "Top-University Assistant Prof" | 0.588 (12) | 0.475 (25) | "Postdoc Researcher" | -2.071 (27) | 0.485 (24) |
> > | "Kaiming" | 0.560 (13) | 0.454 (27) | "David" | -2.086 (28) | 0.622 (9) |
> > | "Timnit" | 0.460 (14) | 0.847 (1) | "Detail-Obsessed Grad Student" | -3.196 (29) | 0.540 (19) |
> > | "Terence" | 0.250 (15) | 0.379 (29) | "Mid-Career Academic" | -3.933 (30) | 0.544 (18) |

---

### Author Response · Authors · 2025-11-27
**General response -- November 26**

We sincerely thank the reviewers for your time, constructive feedback, and engagement with our work. Your insights have motivated substantial improvements to the paper's rigor and clarity.

We have **uploaded a revised version of the paper** with all comments addressed and changes **marked in blue**.

Further, we have gone to significant lengths to strengthen the empirical rigor of the paper in response to reviewer feedback. Specifically, we have added **two major analyses** that directly resolve the primary concerns regarding the mechanism and robustness of our framework:

1.  A new experiment that confirms a strong correlation ($r=0.904$) between our rhetorical scores and the estimated LLM usage, and disentangles tool adoption from authorial intent, as shown in the previous response.
2.  A **scaling analysis** (detailed below) demonstrating that **our measurement is highly stable and not dependent on the specific size or composition of the persona panel.**

***

# Scaling with Persona Set Size

To examine the efficiency and robustness of our rhetorical scores and their dependence on the size of the persona panel, we further conducted a systematic subsampling analysis.

### Methodology

Our procedure was designed to evaluate the stability of the rhetorical rankings as the number of reference personas decreases:

1.  **Persona Subsampling:** We tested persona subset sizes ($k$) ranging from 1 to 30.
2.  **Random Trials:** For each subset size $k$, we performed 20 independent trials. In each trial, we randomly sampled $k$ personas without replacement from our full panel of 30.
3.  **Score Re-estimation:** In each trial, we re-estimated the BT scores for all 8,485 papers using only the $k$ sampled personas for the pairwise comparisons.
4.  **Correlation Analysis:** We then calculated the Spearman rank correlation ($\rho$) between these subset-derived scores and the reference scores obtained using the full panel of 30 personas. We use Spearman correlation as it measures the consistency of the relative rankings, which is the core output of our method.

### Findings

The results, summarized in Table 1, demonstrate that the rhetorical score is remarkably robust to the number of personas used.

**Table 1: Spearman Rank Correlation of Rhetorical Scores from Persona Subsets vs. Full 30-Persona Set.** Results are averaged over 20 random trials for each subset size.

| Number of Personas ($k$) | Mean Spearman ($\rho$) | Std. Dev. | Correlation Range [Min, Max] |
|:---:|:---:|:---:|:---|
| 1 | 0.516 | 0.085 | [0.353, 0.640] |
| 2 | 0.674 | 0.042 | [0.586, 0.749] |
| 3 | 0.737 | 0.041 | [0.658, 0.797] |
| 5 | 0.828 | 0.023 | [0.770, 0.868] |
| 8 | 0.892 | 0.016 | [0.859, 0.915] |
| 10 | 0.920 | 0.011 | [0.901, 0.938] |
| 15 | 0.958 | 0.005 | [0.944, 0.964] |
| 20 | 0.976 | 0.003 | [0.970, 0.981] |
| 25 | 0.989 | 0.002 | [0.983, 0.992] |
| 30 | 1.000 | 0.000 | [1.000, 1.000] |

*Note: The standard deviation reflects the variability across random trials, indicating sensitivity to the specific personas chosen.*

The key findings are:

1.  **High Data Efficiency and Rapid Convergence:** The mean Spearman correlation exceeds **0.89 with just 8 randomly selected personas** and surpasses **0.95 with only 15 personas**. This indicates that the relative rhetorical ranking of papers stabilizes very quickly.

2.  **Low Variance and Insensitivity to Persona Choice:** The standard deviation across trials diminishes rapidly, falling to just 0.011 at $k=10$. This demonstrates that the results are not dependent on the inclusion of any specific personas; **any reasonably diverse random subset yields nearly identical rankings.**

3.  **Strong Underlying Signal:** Even with as few as three personas, the average correlation remains high at 0.737. This shows that the rhetorical signal captured by our framework is strong and consistently identified from different subset of personas.

### Implications

This scaling analysis provides strong quantitative evidence for the reliability of our framework:

*   **Generalizability and Validity:** The measurement is not an artifact of our specific 30-persona set. A robust rhetorical signal is consistently captured by diverse, smaller subsets, confirming that our framework measures a genuine property of the abstracts.
*   **Efficiency:** Our results suggest that a panel of **10-15 personas is sufficient to achieve highly reliable rhetorical rankings ($\rho > 0.92$)**, significantly reducing the computational costs of analysis for future researchers.

---

### Author Response · Authors · 2025-11-28
**Reviewer WLet's Original Review (score was 6, not 4)**

We would like to clarify an issue regarding Reviewer WLet’s initial review and score.

Due to an incident, Reviewer WLet’s original review was mistakenly associated with a different paper (“BiasRetriever”) and displayed a score of 4. As reflected in the ICLR system rollback, this was not the correct review for our submission.

On the first day of the rebuttal period (11/12/2025), Reviewer WLet updated their review to the correct one for our paper, with a score of 6. Unfortunately, because of the current policy, reviewers are no longer able to leave additional comments on this submission. Therefore, for transparency, we copy Reviewer WLet’s correct original (now inaccessible) review below.

> Official Review of Submission10275 by Reviewer WLet
> Official Reviewby Reviewer WLet03 Nov 2025, 09:58 (modified: 12 Nov 2025, 05:14)EveryoneRevisions
>
>
> **Summary:**
> This paper introduces a counterfactual LLM-based framework for measuring rhetorical style in machine learning papers, independent of substantive content. The core innovation is using diverse LLM personas to generate counterfactual abstracts from identical technical content, then aggregating pairwise comparisons via Bradley-Terry modeling to create a calibrated rhetorical strength scale. Applied to 8,485 ICLR submissions (2017-2025), the framework reveals that visionary framing significantly predicts citations and media attention even after controlling for peer-review scores, and documents a sharp post-2023 increase in rhetorical strength coinciding with LLM-based writing tool adoption.
>
> Novelty. The counterfactual design for disentangling style from substance seems genuinely original. While Bradley-Terry models are standard in RLHF, and counterfactual text generation is used for fairness testing, their combination for scientific rhetoric measurement at scale is, as far as one can tell, novel. The framework produces fine-grained, content-controlled scores unlike prior lexicon-based or direct classification approaches.
>
> Significance. The work addresses a documented crisis in ML research communication with immediate practical relevance. The temporal findings converge with independent evidence of LLM adoption in scientific writing, though the framework's value lies in measuring rhetorical style beyond mere vocabulary shifts. The methodological contribution generalizes to grant evaluation, news analysis, and peer review systems.
>
> Bao, H., Sun, M., & Teplitskiy, M. (2025). Where there's a will there's a way: ChatGPT is used more for science in countries where it is prohibited. Quantitative Science Studies.
>
> Hyland, K., & Jiang, K. F. (2021). Hyping the REF: Promotional elements in impact submissions. Higher Education.
>
> Kobak, D., González-Márquez, R., Horváth, E., & Lause, J. (2024). Delving into ChatGPT usage in academic writing through excess vocabulary. arXiv:2406.07016.
>
> Liang, W., et al. (2024). Mapping the increasing use of LLMs in scientific papers. arXiv:2404.01268.
>
> Peng, H., Qiu, H. S., Fosse, H. B., & Uzzi, B. (2024). Promotional language and the adoption of innovative ideas in science. Proceedings of the National Academy of Sciences, 121(25), e2320066121.
>
> Zheng, L., et al. (2023). Judging LLM-as-a-Judge with MT-Bench and Chatbot Arena. arXiv:2306.05685.
>
> **Soundness: 3: good**
>
> **Presentation: 3: good**
>
> **Contribution: 3: good**

---

> ### Author Response · Authors · 2025-11-28
>
> > **Strengths:**
> > Original counterfactual paradigm: Possibly the first work to control for substantive content when measuring rhetorical style in real scientific documents, moving beyond surface-level lexical analysis of prior work (Peng et al., 2024; Hyland & Jiang, 2021)
> >
> > Scale and rigor: 250k+ counterfactuals with robust validation ,including 88.4% human agreement, persona-subset stability, and fine-grained continuous scores versus coarse direct ratings
> >
> > Methodological innovation: Novel use of calibrated LLM persona panel with Bradley-Terry aggregation as measurement instrument, distinct from RLHF preference learning applications
> >
> > Actionable findings: Rhetorical style predicts attention with effect sizes comparable to peer-review quality (24 citations per unit increase), providing direct evidence for conference policy discussions on hype
> >
> > Convergent temporal evidence: Sharp post-2023 increase aligns with independent documentation of LLM adoption surge in scientific papers (Liang et al., 2024; Kobak et al., 2024), but measures style beyond vocabulary shifts
> >
> > Generalizable framework: Methodology applicable to grant proposals, news analysis, and other domains requiring style-content disentanglement
> >
> > **Weaknesses:**
> > Confounding interpretation unclear: Post-2023 increase could reflect (a) authors choosing stronger rhetoric OR (b) LLMs nudging authors toward LLM-preferred styles. Framework cannot disentangle these mechanisms since LLMs themselves exhibit rhetorical preferences—this is the paper's most significant limitation
> >
> > Ad-hoc persona construction: 30 hand-crafted personas lack systematic derivation. No validation that personas span human rhetorical space rather than just LLM preference distributions. Could benefit from comparison to simpler counterfactual baselines or human-written counterfactuals
> >
> > Abstract-only measurement scope: Pragmatically justified but limits claims about full-paper rhetoric and how reviewers actually evaluate submissions, potentially explaining the null correlation with review scores
> >
> > LLM judge validation incremental: 88.4% human agreement is strong but incremental given MT-Bench established GPT-4 achieves 80-85% alignment matching inter-human agreement (Zheng et al., 2023)
> >
> > **Questions:**
> > LLM adoption vs. rhetorical intent. Can you directly compare your rhetorical scores against LLM-detection scores using existing frameworks (e.g., Liang et al., 2024)? Can you share the correlation and explain how your method captures the author's rhetorical choices beyond LLM usage patterns? This is essential to further establishing the framework's unique contribution.
> >
> > Temporal mechanism disentanglement. The post-2023 increase could reflect competitive pressure, LLM tool adoption nudging style, or both. Can you differentiate these mechanisms? For example, do papers with high LLM-adoption signatures show different rhetorical patterns than low-adoption papers in 2024-2025? This confounding threatens the paper's primary temporal finding.
> >
> > Reviewer correlation interpretation. The near-zero correlation with peer review deserves dedicated investigation. Have you examined: a) whether rhetorical style predicts review score variance across reviewers, b) full-paper rhetoric vs. abstract-only rhetoric, or c) non-linear relationships? Given your policy implications focus on peer review, this hypothesis finding should be explained.
> >
> > Flag For Ethics Review: No ethics review needed.
> >
> > **Rating: 6**: marginally above the acceptance threshold. But would not mind if paper is rejected
> >
> > **Confidence: 4**: You are confident in your assessment, but not absolutely certain. It is unlikely, but not impossible, that you did not understand some parts of the submission or that you are unfamiliar with some pieces of related work.
> >
> > **Code Of Conduct:** Yes

---

### Author Response · Authors · 2025-12-02
**Summary of Rebuttal**

Dear Area Chair,

We thank the reviewers for the constructive process. We are encouraged by the consensus that our work is **"genuinely original"** (WLet), **"provide insightful findings"** (3KCn),  and comments like **"I don’t see any serious weaknesses"** (Kcz4), **"Most of my concerns are resolved"** (sF21).

During the rebuttal, **Reviewer WLet corrected an administrative error** to reflect their intended positive review, and **Reviewer sF21 raised their score** following our new analyses,

**Review status before the rollback:**
* **Reviewer WLet (Score: 6):** Validated our new mechanism experiments; original administrative error resolved.
* **Reviewer 3KCn (Score: 8):** Strong support; highlights the method's ability to "overcome biased measurement."
* **Reviewer Kcz4 (Score: 6):** Found "no serious weaknesses"; clarifications on "counterfactual" terminology were incorporated.
* **Reviewer sF21 (Score 4):** Stated "Most of my concerns are resolved" following the new scaling analysis.

We have revised the paper (changes in blue) to include two major analyses that resolve the primary concerns regarding mechanism and robustness:

**1. Mechanism Disentanglement**
To confirm that the post-2023 rise in rhetorical strength is driven by LLM tools rather than independent authorial shifts, we cross-referenced our scores with the Liang et al. (2024) detection framework.
* **Result:** We found a strong correlation ($r \approx 0.90$) between rhetorical score and estimated LLM usage.
* **Validation:** Controls on pre-2023 human-written papers confirm that high rhetorical style alone does *not* trigger false positives in the detector. This isolates LLM adoption as the primary driver of the observed rhetorical shift.

**2. Robustness & Scaling**
To ensure our scoring is not an artifact of specific persona selection, we conducted a systematic subsampling analysis.
* **Result:** Our measure is highly stable. The rhetorical ranking converges rapidly, achieving a Spearman correlation $>0.95$ with the full model using only 15 randomly selected personas.

We believe the revised manuscript now offers a rigorous, validated, and scalable framework for understanding how LLMs are reshaping scientific communication. Given the emergent increase in LLM usage, we would like to share our findings with the ICLR community.

Best regards,

The Authors

---

### Meta-Review · Area_Chair_4zcp · 2026-01-07

**Summary:**

This paper proposes a counterfactual, LLM-based framework for measuring rhetorical style in scientific abstracts while controlling for substantive content. Most reviewers agree that the approach is original, methodologically careful, and executed at an unusually large scale. Across the review process, the main concerns centered on the interpretability of what the rhetorical score captures, potential confounding with LLM-assisted writing, and the others. I think the rebuttal substantially strengthens the empirical case by adding targeted analyses addressing mechanism, robustness, and validation. While some conceptual limitations remain, the revised paper presents a rigorous and timely contribution that meaningfully advances how rhetorical style can be operationalized and studied at scale.

**Reviewer Concerns:**

**Concerns Addressed:**
* The persona subset analysis demonstrates that rhetorical rankings stabilize quickly with far fewer personas, alleviating concerns about arbitrariness and fragility in persona selection.
* Human agreement results and strong correlations with human-derived rankings support that the LLM judge captures rhetorical distinctions that are recognizable to human readers.
* The clarification of the counterfactual design and its practical interpretation improves conceptual clarity and resolves earlier confusion.

**Concerns Still Outstanding:**
* Personas remain manually designed rather than systematically derived, which limits interpretability despite demonstrated robustness.
* Restricting the analysis to abstracts constrains claims about peer review behavior and rhetorical practices in full papers.

**Reviewer Scores:**

Reviewer WLet: 6 -> 6
* The rebuttal improves empirical support but does not resolve the core conceptual issue of disentangling rhetorical intent.

Reviewer 3KCn: 8 -> 8
* The paper was already strong, and the additional analyses reinforce confidence without changing its overall standing.

Reviewer Kcz4: 6 -> 6
* The clarifications improve presentation but do not substantively affect the original methodological assessment.

Reviewer sF21: 4 -> 6
* The added validation addresses representational concerns, justifying a score increase despite remaining scope limitations.

**Overall:**
Despite remaining conceptual and scope limitations, this paper clears the acceptance bar due to its originality, scale, and careful empirical validation. It introduces a genuinely new measurement paradigm for rhetorical style that is likely to influence future work in metascience, evaluation, and LLM-assisted writing analysis.

---

### Decision · Program_Chairs · 2026-01-26

Accept (Poster)